# Computation-Aware Gaussian Processes: Model Selection And Linear-Time Inference

**Jonathan Wenger**[1]    **Kaiwen Wu**[2]    **Philipp Hennig**[3]    **Jacob R. Gardner**[2]

**Geoff Pleiss**[4]    **John P. Cunningham**[1]

[1] Columbia University
[2] University of Pennsylvania
[3] University of Tübingen, Tübingen AI Center
[4] University of British Columbia, Vector Institute

## Abstract

Model selection in Gaussian processes scales prohibitively with the size of the training dataset, both in time and memory. While many approximations exist, all incur inevitable approximation error. Recent work accounts for this error in the form of computational uncertainty, which enables—at the cost of quadratic complexity—an *explicit* tradeoff between computational efficiency and precision. Here we extend this development to model selection, which requires significant enhancements to the existing approach, including linear-time scaling in the size of the dataset. We propose a novel training loss for hyperparameter optimization and demonstrate empirically that the resulting method can outperform SGPR, CGGP and SVGP, state-of-the-art methods for GP model selection, on medium to large-scale datasets. Our experiments show that model selection for computation-aware GPs trained on 1.8 million data points can be done within a few hours on a single GPU. As a result of this work, Gaussian processes can be trained on large-scale datasets without significantly compromising their ability to quantify uncertainty—a fundamental prerequisite for optimal decision-making.

## 1   Introduction

Gaussian Processes (GPs) remain a popular probabilistic model class, despite the challenges in scaling them to large datasets. Since both computational and memory resources are limited in practice, approximations are necessary for both inference and model selection. Among the many approximation methods, perhaps the most common approach is to map the data to a lower-dimensional representation. The resulting posterior approximations typically have a functional form similar to the exact GP posterior, except where posterior mean and covariance feature *low-rank updates*. This strategy can be explicit—by either defining feature functions (e.g. Nyström [1], RFF [2])—or a lower-dimensional latent inducing point space (e.g. SoR, DTC, FITC [3], SGPR [4], SVGP [5]), or implicit—by using an iterative numerical method (e.g. CGGP [6–10]). All of these methods then compute coefficients for this lower-dimensional representation from the full set of observations by direct projection (e.g. CGGP) or via an optimization objective (e.g. SGPR, SVGP).

While effective and widely used in practice, the inevitable approximation error adversely impacts predictions, uncertainty quantification, and ultimately downstream decision-making. Many proposed methods come with theoretical error bounds [e.g. 2, 11–14], offering insights into the scaling and asymptotic properties of each method. However, theoretical bounds often require too many assumptions about the data-generating process to offer "real-world" guarantees [15], and in practice, the fidelity of the approximation is ultimately determined by the available computational resources.

38th Conference on Neural Information Processing Systems (NeurIPS 2024).

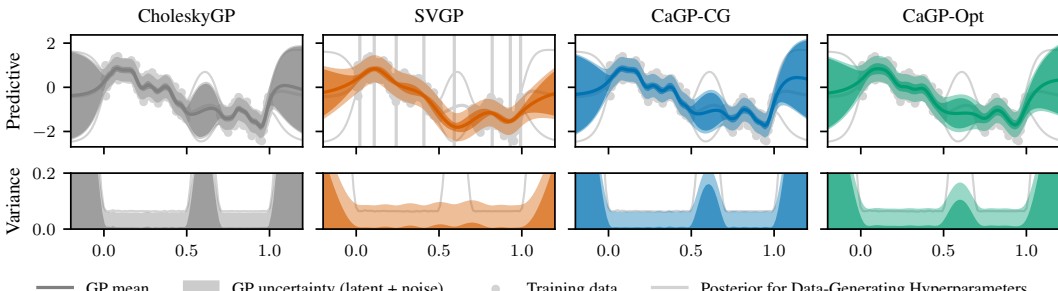

Figure 1: Comparison of an exact GP posterior (CholeskyGP) and three scalable approximations: SVGP, CaGP-CG and CaGP-Opt (ours). Hyperparameters for each model were optimized using model selection strategies specific to each approximation. The posterior predictive given the data-generating hyperparameters is denoted by gray lines and for each method the posterior (dark-shaded) and the posterior predictive are shown (light-shaded). While all methods, including the exact GP, do not recover the data-generating process, CaGP-CG and CaGP-Opt are much closer than SVGP. SVGP expresses almost no posterior variance near the inducing point in the data-sparse region and thus almost all deviation from the posterior mean is considered to be observational noise. In contrast, CaGP-CG and CaGP-Opt express significant posterior variance in regions with no data.

One central pathology is overconfidence, which has been shown to be detrimental in key applications of GPs such as Bayesian optimization [e.g. variance starvation of RFF, 16], and manifests itself even in state-of-the-art variational methods like SVGP. SVGP, because it treats inducing variables as "virtual observations", can be overconfident at the locations of the inducing points if they are not in close proximity to training data, which becomes increasingly likely in higher dimensions. This phenomenon can be seen in a toy example in Figure 1, where SVGP has near zero posterior variance at the inducing point away from the data. See also Section S5.1 for a more detailed analysis.

These approximation errors are a central issue in inference, but they are exacerbated in model selection, where errors compound and result in biased selections of hyperparameters [12, 17, 18]. Continuing the example, SVGP has been observed to overestimate the observation noise [18], which can lead to oversmoothing. This issue can also be seen in Figure 1, where the SVGP model produces a smoother posterior mean than the exact (Cholesky)GP and attributes most variation from the posterior mean to observational noise (see also Figure S3(b)). There have been efforts to understand these biases [18] and to mitigate the impact of approximation error on model selection for certain approximations [e.g. CGGP, 12], but overcoming these issues for SVGP remains a challenge.

Recently, Wenger et al. [19] introduced computation-aware Gaussian processes (CaGP), a class of GP approximation methods which—for a fixed set of hyperparameters—provably does not suffer from overconfidence. Like SVGP and the other approximations mentioned above, CaGP also relies on low-rank posterior updates. Unlike these other methods, however, CaGP's posterior updates are constructed to guarantee that its posterior variance is always larger than the exact GP variance. This conservative estimate can be interpreted as additional uncertainty quantifying the approximation error due to limited computation; i.e. *computational uncertainty*. However, so far CaGP has fallen short in demonstrating wallclock time improvements for posterior inference over variational methods and model selection has so far remained an open problem.

**Contributions** In this work, we extend computation-aware Gaussian processes by demonstrating how to perform inference in linear time in the number of training data, while maintaining its theoretical guarantees. Second, we propose a novel objective that allows model selection without a significant bias that would arise from naively conducting model selection on the projected GP. In detail, we enforce a sparsity constraint on the "actions" of the method, which unlocks linear-time inference, in a way that is amenable to hardware acceleration. We optimize these actions end-to-end alongside the hyperparameters with respect to a custom training loss, to optimally retain as much information from the data as possible given a limited computational budget. The resulting hyperparameters are less prone to oversmoothing and attributing variation to observational noise, as can be seen in Figure 1, when compared to SVGP. We demonstrate that our approach is strongly competitive on large-scale data with state-of-the-art variational methods, such as SVGP, without inheriting their pathologies. As a consequence of our work, one can train GPs on up to 1.8 million data points in a few hours on a single GPU without adversely impacting uncertainty quantification.

## 2 Background

We aim to learn a latent function mapping from $\mathbb{X} \subseteq \mathbb{R}^d$ to $\mathbb{Y} \subseteq \mathbb{R}$ given a training dataset $\boldsymbol{X} = (\boldsymbol{x}_1, \dots, \boldsymbol{x}_n) \in \mathbb{R}^{n \times d}$ of $n$ inputs $\boldsymbol{x}_j \in \mathbb{R}^d$ and corresponding targets $\boldsymbol{y} = (y_1, \dots, y_n)^\mathsf{T} \in \mathbb{R}^n$.

**Gaussian Processes**  A *Gaussian process* $f \sim \mathcal{GP}(\mu, K_{\boldsymbol{\theta}})$ is a stochastic process with mean function $\mu$ and kernel $K_{\boldsymbol{\theta}}$ such that $\boldsymbol{f} = f(\boldsymbol{X}) = (f(\boldsymbol{x}_1), \dots, f(\boldsymbol{x}_n))^\mathsf{T} \sim \mathcal{N}(\boldsymbol{\mu}, \boldsymbol{K}_{\boldsymbol{\theta}})$ is jointly Gaussian with mean $\boldsymbol{\mu}_i = \mu(\boldsymbol{x}_i)$ and covariance $\boldsymbol{K}_{ij} = K_{\boldsymbol{\theta}}(\boldsymbol{x}_i, \boldsymbol{x}_j)$. The kernel $K_{\boldsymbol{\theta}}$ depends on hyperparameters $\boldsymbol{\theta} \in \mathbb{R}^p$, which we omit in our notation. Assuming $\boldsymbol{y} \mid f(\boldsymbol{X}) \sim \mathcal{N}(f(\boldsymbol{X}), \sigma^2 \boldsymbol{I})$, the posterior is a Gaussian process $\mathcal{GP}(\mu_\star, K_\star)$ where the mean and covariance functions evaluated at a test input $\boldsymbol{x}_\diamond \in \mathbb{R}^d$ are given by

$$
\begin{aligned}
\mu_\star(f(\boldsymbol{x}_\diamond)) &= \mu(\boldsymbol{x}_\diamond) + K(\boldsymbol{x}_\diamond, \boldsymbol{X})\boldsymbol{v}_\star, \\
K_\star(f(\boldsymbol{x}_\diamond), f(\boldsymbol{x}_\diamond)) &= K(\boldsymbol{x}_\diamond, \boldsymbol{x}_\diamond) - K(\boldsymbol{x}_\diamond, \boldsymbol{X})\hat{\boldsymbol{K}}^{-1} K(\boldsymbol{X}, \boldsymbol{x}_\diamond),
\end{aligned}
\tag{1}
$$

where $\hat{\boldsymbol{K}} = \boldsymbol{K} + \sigma^2 \boldsymbol{I}$ and the *representer weights* are defined as $\boldsymbol{v}_\star = \hat{\boldsymbol{K}}^{-1}(\boldsymbol{y} - \boldsymbol{\mu})$.

In model selection, the computational bottleneck when optimizing kernel hyperparameters $\boldsymbol{\theta}$ is the repeated evaluation of the *negative* log-*marginal likelihood*

$$
\ell^{\mathrm{NLL}}(\boldsymbol{\theta}) = -\log p(\boldsymbol{y} \mid \boldsymbol{\theta}) = \tfrac{1}{2}\big(\underbrace{(\boldsymbol{y} - \boldsymbol{\mu})^\mathsf{T} \hat{\boldsymbol{K}}^{-1}(\boldsymbol{y} - \boldsymbol{\mu})}_{\text{quadratic loss: promotes data fit}} + \underbrace{\log\det(\hat{\boldsymbol{K}})}_{\text{model complexity}} + n\log(2\pi)\big)
\tag{2}
$$

and its gradient. Computing (2) and its gradient via a Cholesky decomposition has time complexity $\mathcal{O}(n^3)$ and requires $\mathcal{O}(n^2)$ memory, which is prohibitive for large $n$.

**Sparse Gaussian Process Regression (SGPR) [4]**  Given a set of $m \ll n$ *inducing points* $\boldsymbol{Z} = (\boldsymbol{z}_1, \dots, \boldsymbol{z}_m)^\mathsf{T}$ and defining $\boldsymbol{u} := f(\boldsymbol{Z}) = (f(\boldsymbol{z}_1), \dots, f(\boldsymbol{z}_m))^\mathsf{T}$, SGPR defines a variational approximation to the posterior through the factorization $p_\star(f(\cdot) \mid \boldsymbol{y}) \approx q(f(\cdot)) = \int p(f(\cdot) \mid \boldsymbol{u})\, q(\boldsymbol{u}) d\boldsymbol{u}$, where $q(\boldsymbol{u})$ is an $m$-dimensional multivariate Gaussian. The mean and covariance of $q(\boldsymbol{u})$ (denoted as $\boldsymbol{m} := \mathbb{E}_q(\boldsymbol{u}), \boldsymbol{\Sigma} := \mathrm{Cov}_q(\boldsymbol{u})$) are jointly optimized alongside the kernel hyperparameters $\boldsymbol{\theta}$ using the evidence lower bound (ELBO) as an objective:

$$
\boldsymbol{m}, \boldsymbol{\Sigma}, \boldsymbol{\theta}, \boldsymbol{Z} = \arg\min_{\boldsymbol{m}, \boldsymbol{\Sigma}, \boldsymbol{\theta}, \boldsymbol{Z}} \ell^{\mathrm{ELBO}},
\tag{3}
$$

$$
\begin{aligned}
\ell^{\mathrm{ELBO}}(\boldsymbol{m}, \boldsymbol{\Sigma}, \boldsymbol{\theta}, \boldsymbol{Z}) &:= \ell^{\mathrm{NLL}}(\boldsymbol{\theta}) + \mathrm{KL}(q(\boldsymbol{f}) \,\|\, p_\star(\boldsymbol{f} \mid \boldsymbol{y}, \boldsymbol{\theta})) \\
&= -\mathbb{E}_{q(\boldsymbol{f})}(\log p(\boldsymbol{y} \mid \boldsymbol{f})) + \mathrm{KL}(q(\boldsymbol{u}) \,\|\, p(\boldsymbol{u})) \geq -\log p(\boldsymbol{y} \mid \boldsymbol{\theta}).
\end{aligned}
\tag{4}
$$

The inducing point locations $\boldsymbol{Z}$ can be either optimized as additional parameters during training or chosen a-priori, typically in a data-dependent way [see e.g. Sec. 7.2 of 20]. Following Titsias [4], ELBO optimization and posterior inference both require $\mathcal{O}(nm^2)$ computation and $\mathcal{O}(nm)$ memory, a significant improvement over the costs of exact GPs.

**Stochastic Variational Gaussian Processes (SVGP) [5]**  SVGP extends SGPR to reduce complexity further to $\mathcal{O}(m^3)$ computation and $\mathcal{O}(m^2)$ memory. It accomplishes this reduction by replacing the first term in Equation (4) with an unbiased approximation

$$
\mathbb{E}_{q(\boldsymbol{f})}(\log p(\boldsymbol{y} \mid \boldsymbol{f})) = \mathbb{E}_{q(\boldsymbol{f})}\Big(\sum_{i=1}^n \log p(y_i \mid f(\boldsymbol{x}_i))\Big) \approx n\, \mathbb{E}_{q(f(\boldsymbol{x}_i))}(\log p(y_i \mid f(\boldsymbol{x}_i))).
$$

Following Hensman et al. [5], we optimize $\boldsymbol{m}, \boldsymbol{\Sigma}$ alongside $\boldsymbol{\theta}, \boldsymbol{Z}$ through joint gradient updates. Because the asymptotic complexities no longer depend on $n$, SVGP can scale to extremely large datasets that would not be able to fit into computer/GPU memory.

**Computation-aware Gaussian Process Inference (CaGP) [19]**  CaGP[1] maps the data $\boldsymbol{y}$ into a lower dimensional subspace defined by its *actions* $\boldsymbol{S}_i \in \mathbb{R}^{n \times i}$ on the data, which defines an approximate posterior $\mathcal{GP}(\mu_i, K_i)$ with

$$
\begin{aligned}
\mu_i(\boldsymbol{x}_\diamond) &= \mu(\boldsymbol{x}_\diamond) + K(\boldsymbol{x}_\diamond, \boldsymbol{X})\boldsymbol{v}_i \\
K_i(\boldsymbol{x}_\diamond, \boldsymbol{x}_\diamond) &= K(\boldsymbol{x}_\diamond, \boldsymbol{x}_\diamond) - K(\boldsymbol{x}_\diamond, \boldsymbol{X})C_i K(\boldsymbol{X}, \boldsymbol{x}_\diamond),
\end{aligned}
\tag{5}
$$

---

[1]Wenger et al. [19] named their computation-aware inference algorithm "IterGP", to emphasize its iterative nature (see Algorithm S1). We adopt the naming "CaGP" instead, since if the actions $\boldsymbol{S}$ are not chosen sequentially, it is more efficient to compute the computation-aware posterior non-iteratively (see Algorithm S2).

where $\boldsymbol{C}_i = \boldsymbol{S}_i(\boldsymbol{S}_i^{\mathsf{T}}\hat{\boldsymbol{K}}\boldsymbol{S}_i)^{-1}\boldsymbol{S}_i^{\mathsf{T}} \approx \boldsymbol{K}^{-1}$ is a low-rank approximation of the precision matrix and $\boldsymbol{v}_i = \boldsymbol{C}_i(\boldsymbol{y} - \boldsymbol{\mu}) \approx \boldsymbol{v}_\star$ approximates the representer weights. Both converge to the corresponding exact quantity as the number of iterations, equivalently the downdate rank, $i \to n$. Note that the CaGP posterior only depends on the space spanned by the columns of $\boldsymbol{S}_i$, not the actual matrix (Lemma S4). Finally, the CaGP posterior can be computed in $\mathcal{O}(n^2 i)$ time and $\mathcal{O}(ni)$ memory.

CaGP captures the approximation error incurred by limited computational resources as additional uncertainty about the latent function. More precisely, for any data-generating function $y \in \mathbb{H}_{K^\sigma}$ in the RKHS $\mathbb{H}_{K^\sigma}$ defined by the kernel, the worst-case squared error of the corresponding approximate posterior mean $\mu_i^y$ is equal to the approximate predictive variance (see Theorem S2):

$$\sup_{y \in \mathbb{H}_{K^\sigma}, \|y\|_{\mathbb{H}_{K^\sigma}} \leq 1} \left(y(\boldsymbol{x}_\diamond) - \mu_i^y(\boldsymbol{x}_\diamond)\right)^2 = K_i(\boldsymbol{x}_\diamond, \boldsymbol{x}_\diamond) + \sigma^2 \tag{6}$$

This guarantee is identical to one for the exact GP posterior mean and variance (see Theorem S1), except with the *approximate* quantities instead.[2] Additionally, it holds that CaGP's marginal (predictive) variance is always larger or equal to the (predictive) variance of the exact GP and monotonically decreasing, i.e. $K_i(\boldsymbol{x}_\diamond, \boldsymbol{x}_\diamond) \geq K_j(\boldsymbol{x}_\diamond, \boldsymbol{x}_\diamond) \geq K_n(\boldsymbol{x}_\diamond, \boldsymbol{x}_\diamond) = K_\star(\boldsymbol{x}_\diamond, \boldsymbol{x}_\diamond)$ for $i \leq j \leq n$ (see Proposition S1). Therefore, given fixed hyperparameters, CaGP is guaranteed to *never* be overconfident and as the computational budget increases, the precision of its estimate increases. Here we call such a posterior *computation-aware*, and we will extend the use of this object to model selection.

## 3 Model Selection in Computation-Aware Gaussian Processes

Model selection for GPs most commonly entails maximizing the evidence $\log p(\boldsymbol{y} \mid \boldsymbol{\theta})$ as a function of the kernel hyperparameters $\boldsymbol{\theta} \in \mathbb{R}^p$. As with posterior inference, evaluating this objective and its gradient is computationally prohibitive in the large-scale setting. Therefore, our central goal will be to perform model selection for computation-aware Gaussian processes in order to scale to a large number of training data while avoiding the introduction of pathologies via approximation.

We begin by viewing the computation-aware posterior as *exact inference* assuming we can only observe $i$ linear projections $\tilde{\boldsymbol{y}}$ of the data defined by (linearly independent) actions $\boldsymbol{S}_i \in \mathbb{R}^{n \times i}$, i.e.

$$\tilde{\boldsymbol{y}} := \boldsymbol{S}_i'^{\mathsf{T}}\boldsymbol{y} \in \mathbb{R}^i, \qquad \text{where} \qquad \boldsymbol{S}_i' = \boldsymbol{S}_i \, \text{chol}(\boldsymbol{S}_i^{\mathsf{T}}\boldsymbol{S}_i)^{-\mathsf{T}} \in \mathbb{R}^{n \times i} \tag{7}$$

is the action matrix with orthonormalized columns. The corresponding likelihood is given by

$$p(\tilde{\boldsymbol{y}} \mid f(\boldsymbol{X})) = \mathcal{N}\left(\tilde{\boldsymbol{y}}; \boldsymbol{S}_i'^{\mathsf{T}}f(\boldsymbol{X}), \sigma^2\boldsymbol{I}\right). \tag{8}$$

As we show in Lemma S1, the resulting Bayesian posterior is then given by Equation (5). Recall that the CaGP posterior only depends on the column space of the actions $\boldsymbol{S}_i$ (see Lemma S4), which is why Equation (5) can be written in terms of $\boldsymbol{S}_i$ directly rather than using $\boldsymbol{S}_i'$.[3]

**Projected-Data Log Marginal Likelihood** The reinterpretation of the computation-aware posterior as exact Bayesian inference immediately suggests using evidence maximization for the projected data $\tilde{\boldsymbol{y}} = \boldsymbol{S}_i'^{\mathsf{T}}\boldsymbol{y} \in \mathbb{R}^i$ as the model selection criterion, leading to the following loss (see Lemma S2):

$$\ell_{\text{proj}}^{\text{NLL}}(\boldsymbol{\theta}) = -\log p(\tilde{\boldsymbol{y}} \mid \boldsymbol{\theta}) = -\log \mathcal{N}\left(\tilde{\boldsymbol{y}}; \boldsymbol{S}_i'^{\mathsf{T}}\boldsymbol{\mu}, \boldsymbol{S}_i'^{\mathsf{T}}\hat{\boldsymbol{K}}\boldsymbol{S}_i'\right)$$

$$= \frac{1}{2}\Big(\underbrace{(\boldsymbol{y} - \boldsymbol{\mu})^{\mathsf{T}}\boldsymbol{S}_i(\boldsymbol{S}_i^{\mathsf{T}}\hat{\boldsymbol{K}}\boldsymbol{S}_i)^{-1}\boldsymbol{S}_i^{\mathsf{T}}(\boldsymbol{y} - \boldsymbol{\mu})}_{\text{quadratic loss: promotes fitting projected data } \tilde{\boldsymbol{y}}} + \underbrace{\log\det(\boldsymbol{S}_i^{\mathsf{T}}\hat{\boldsymbol{K}}\boldsymbol{S}_i) - \overbrace{\log\det(\boldsymbol{S}_i^{\mathsf{T}}\boldsymbol{S}_i)}^{\text{penalizes near-colinear actions}}}_{\text{projected model complexity}} + i\log(2\pi)\Big) \tag{9}$$

Equation (9) involves a Gaussian random variable of dimension $i \ll n$, and so all previously intractable quantities in Equation (2) (i.e. the inverse and determinant) are now cheap to compute. Analogous to the CaGP posterior, we can express the projected-data log marginal likelihood fully in terms of the actions $\boldsymbol{S}_i$ without having to orthonormalize, which results in an additional term penalizing colinearity. Unfortunately, this training loss does not lead to good generalization performance, as there is only training signal in the $i$-dimensional space spanned by the actions $\boldsymbol{S}_i$ the data are projected onto. Specifically, the quadratic loss term only promotes fitting the projected data $\tilde{\boldsymbol{y}}$, not all observations $\boldsymbol{y}$. See Figures S1 and S2 for experimental validation of this claim.

---

[2]At first glance, SVGP satisfies a similar result (Theorem S3). However, this statement does *not* express the variance in terms of the worst-case squared error to the "true" latent function. See Section S1.2 for details.

[3]Given this observation, we sometimes abuse terminology and refer to the actions as "projecting" the data to a lower-dimensional space, although $\boldsymbol{S}_i$ does not need to have orthonormal columns.

**ELBO Training Loss**  Motivated by this observation, we desire a tractable training loss that encourages maximizing the evidence for the *entire set of targets* $\boldsymbol{y}$. Importantly though, we need to accomplish this evidence maximization without incurring prohibitive $\mathcal{O}(n^3)$ computational cost. We define a variational objective, using the computation-aware posterior $q_i(\boldsymbol{f} \mid \boldsymbol{y}, \boldsymbol{\theta})$ in Equation (5) to define a variational family $\mathcal{Q} := \big\{ q_i(\boldsymbol{f}) = \mathcal{N}(\boldsymbol{f}; \mu_i(\boldsymbol{X}), K_i(\boldsymbol{X}, \boldsymbol{X})) \mid \boldsymbol{S} \in \mathbb{R}^{n \times i} \big\}$ parametrized by the action matrix $\boldsymbol{S}$. We can then specify a (negative) evidence lower bound (ELBO) as follows:

$$
\ell_{\text{CaGP}}^{\text{ELBO}}(\boldsymbol{\theta}) = \underbrace{\ell^{\text{NLL}}(\boldsymbol{\theta})}_{\text{balances data fit and model complexity}} + \underbrace{\text{KL}(q_i \parallel p_\star)}_{\text{regularizes s.t. } q_i \approx p_\star} \geq -\log p(\boldsymbol{y} \mid \boldsymbol{\theta}).
\tag{10}
$$

This loss promotes learning the same hyperparameters as if we were to maximize the computationally intractable evidence $\log p(\boldsymbol{y} \mid \boldsymbol{\theta})$ while minimizing the error due to posterior approximation $q_i(\boldsymbol{f}) \approx p_\star(\boldsymbol{f} \mid \boldsymbol{y}, \boldsymbol{\theta})$. In the computation-aware setting, this translates to minimizing computational uncertainty, which captures this inevitable error.

Although both the evidence and KL terms of the ELBO involve computationally intractable terms, these problematic terms cancel out when combined. This results in an objective that costs the same as evaluating CaGP's predictive distribution, i.e.

$$
\ell_{\text{CaGP}}^{\text{ELBO}}(\boldsymbol{\theta}) = \frac{1}{2} \bigg( \frac{1}{\sigma^2} \Big( \|\boldsymbol{y} - \mu_i(\boldsymbol{X})\|_2^2 + \sum_{j=1}^{n} K_i(\boldsymbol{x}_j, \boldsymbol{x}_j) \Big) + (n - i) \log(\sigma^2) + n \log(2\pi)
$$
$$
+ \tilde{\boldsymbol{v}}_i^{\mathsf{T}} \boldsymbol{S}^{\mathsf{T}} \boldsymbol{K} \boldsymbol{S} \tilde{\boldsymbol{v}}_i - \text{tr}((\boldsymbol{S}^{\mathsf{T}} \hat{\boldsymbol{K}} \boldsymbol{S})^{-1} \boldsymbol{S}^{\mathsf{T}} \boldsymbol{K} \boldsymbol{S}) + \log \det(\boldsymbol{S}^{\mathsf{T}} \hat{\boldsymbol{K}} \boldsymbol{S}) - \log \det(\boldsymbol{S}^{\mathsf{T}} \boldsymbol{S}) \bigg)
\tag{11}
$$

where $\tilde{\boldsymbol{v}}_i = (\boldsymbol{S}^{\mathsf{T}} \hat{\boldsymbol{K}} \boldsymbol{S})^{-1} \boldsymbol{S}^{\mathsf{T}}(\boldsymbol{y} - \boldsymbol{\mu})$. For a derivation of this expression of the loss see Lemma S3. If we compare the training loss $\ell_{\text{CaGP}}^{\text{ELBO}}$ in Equation (10) with the projected-data NLL in Equation (9), there is an explicit squared loss penalty on *the entire data* $\boldsymbol{y}$, rather than just for the projected data $\tilde{\boldsymbol{y}}$, resulting in better generalization as Figures S1 and S2 show on synthetic data. In our experiments, this objective was critical to achieving state-of-the-art performance.

## 4  Choice of Actions

So far we have not yet specified the actions $\boldsymbol{S} \in \mathbb{R}^{n \times i}$ mapping the data to a lower-dimensional space. Ideally, we would want to optimally compress the data both for inference and model selection.

**Posterior Entropy Minimization**  We can interpret choosing actions $\boldsymbol{S}$ as a form of active learning, where instead of just observing individual datapoints, we allow ourselves to observe linear combinations of the data $\boldsymbol{y}$. Taking an information-theoretic perspective [21], we would then aim to choose actions such that *uncertainty about the latent function is minimized*. In fact, we show in Lemma S5 that given a prior $f \sim \mathcal{GP}(\mu, K)$ for the latent function and a budget of $i$ actions $\boldsymbol{S}$, the actions that minimize the entropy of the posterior at the training data

$$
(\boldsymbol{s}_1, \ldots, \boldsymbol{s}_i) = \underset{\boldsymbol{S} \in \mathbb{R}^{n \times i}}{\arg \min} \, \text{H}_{p(f(\boldsymbol{X}) \mid \boldsymbol{S}^{\mathsf{T}} \boldsymbol{y})}(f(\boldsymbol{X}))
\tag{12}
$$

are the top-$i$ eigenvectors $\boldsymbol{s}_1, \ldots, \boldsymbol{s}_i$ of $\hat{\boldsymbol{K}}$ in descending order of the eigenvalue magnitude (see also Zhu et al. [22]). Unfortunately, computing these actions is just as prohibitive computationally as computing the intractable GP posterior.

**(Conjugate) Gradient / Residual Actions**  Due to the intractability of choosing actions to minimize posterior entropy, we could try to do so approximately. The Lanczos algorithm [23] is an iterative method to approximate the eigenvalues and eigenvectors of symmetric positive-definite matrices. Given an appropriate seed vector, the space spanned by its approximate eigenvectors is equivalent to the span of the gradients / residuals $\boldsymbol{r}_i = (\boldsymbol{y} - \boldsymbol{\mu}) - \hat{\boldsymbol{K}} \boldsymbol{v}_{i-1}$ of the method of Conjugate Gradients (CG) [24] when used to iteratively compute an approximation $\boldsymbol{v}_i \approx \boldsymbol{v}_\star = \hat{\boldsymbol{K}}^{-1}(\boldsymbol{y} - \boldsymbol{\mu})$ to the representer weights $\boldsymbol{v}_\star$. We show in Lemma S4 that the CaGP posterior only depends on the span of its actions. Therefore choosing approximate eigenvectors computed via the Lanczos process as actions is equivalent to using CG residuals. This allows us to reinterpret CaGP-CG, as introduced by Wenger et al. [19], as approximately minimizing posterior entropy.[4] See Section S3.1 for details.

---

[4]Wenger et al. [Sec. 2.1 of 19] showed that CaGP using CG residuals as actions recovers Conjugate-Gradient-based GPs (CGGP) [7, 9, 10] in its posterior mean, extending this observation to CGGP.

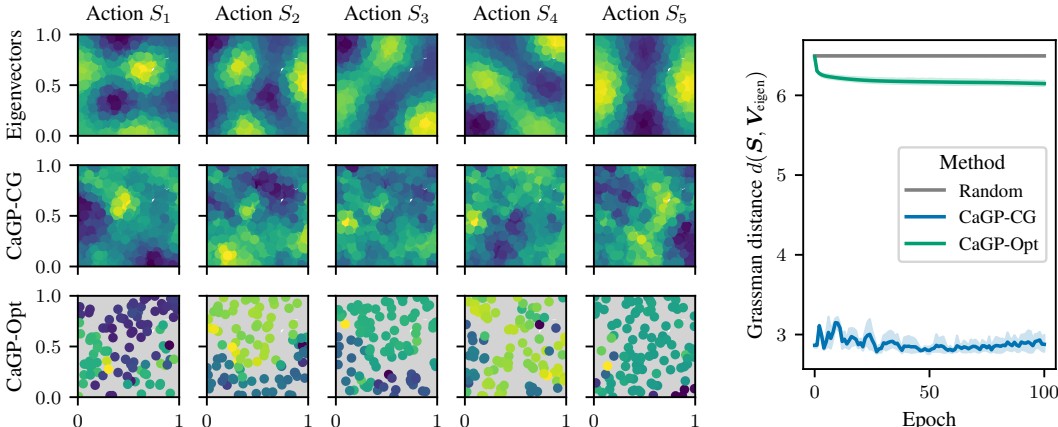

Figure 2: *Visualization of action vectors defining the data projection.* We perform model selection using two CaGP variants, with CG and learned sparse actions—denoted as CaGP-CG, and CaGP-Opt—on a toy 2-dimensional dataset. Left: For each $\boldsymbol{x}_j \in \{\boldsymbol{x}_1, \ldots, \boldsymbol{x}_n\}$, we plot the magnitude of the entries of the top-5 eigenvectors of $\hat{\boldsymbol{K}}$ and of the first five action vectors. Yellow denotes larger magnitudes; blue denotes smaller magnitudes. Right: We compare the span of the actions $\boldsymbol{S}$ against the top-$i$ eigenspace throughout training by measuring the Grassman distance between the two subspaces (see also Section S5.2). CaGP-CG actions are closer to the kernel eigenvectors than the CaGP-Opt actions, both of which are more closely aligned than randomly chosen actions.

As Figure 2 illustrates, CG actions are similar to the top-$i$ eigenspace all throughout hyperparameter optimization. However, this choice of actions focuses exclusively on posterior inference and incurs quadratic time complexity $\mathcal{O}(n^2 i)$.

**Learned Sparse Actions**  So far in our action choices we have entirely ignored model selection and tried to choose optimal actions assuming *fixed* kernel hyperparameters. The second contribution of this work, aside from demonstrating how to perform model selection, is recognizing that the actions should be informed by the outer optimization loop for the hyperparameters. We thus optimize the actions alongside the hyperparameters *end-to-end*, meaning the training loss for model selection defines what data projections are informative. This way the actions are adaptive to the hyperparameters without spending unnecessary budget on computing approximately optimal actions for the current choice of hyperparameters. Specifically, the actions are chosen by optimizing $\ell_{\text{CaGP}}^{\text{ELBO}}$ as a function of the hyperparameters $\boldsymbol{\theta}$ and the actions $\boldsymbol{S}$, s.t.

$$(\boldsymbol{\theta}_\star, \boldsymbol{S}_i) = \arg\min_{(\boldsymbol{\theta}, \boldsymbol{S})} \ell_{\text{CaGP}}^{\text{ELBO}}(\boldsymbol{\theta}, \boldsymbol{S}). \qquad (13)$$

Naively this approach introduces an $n \times i$ dimensional optimization problem, which in general is computationally prohibitive. To keep the computational cost low and to optimally leverage hardware acceleration via GPUs, we impose a sparse block structure on the actions (see Eq. 14) where each block is a column vector $\boldsymbol{s}'_j \in \mathbb{R}^{k \times 1}$ with $k = n/i$ entries such that the total number of trainable action parameters, i.e. non-zero entries $\text{nnz}(\boldsymbol{S}) = k \cdot i = n$, equals the number of training data. Due to the sparsity, these actions cannot perfectly match the maxi-

$$\boldsymbol{S} = \begin{bmatrix} \boldsymbol{s}'_1 & 0 & \cdots & 0 \\ 0 & \boldsymbol{s}'_2 & & 0 \\ \vdots & & \ddots & \vdots \\ 0 & \cdots & 0 & \boldsymbol{s}'_i \end{bmatrix} \qquad (14)$$

mum eigenvector actions. Nevertheless, we see in Figure 2 that optimizing these sparse actions in conjunction with hyperparameter optimization produces a nontrivial alignment with optimal action choice minimizing posterior entropy. Importantly, the sparsity constraint not only reduces the dimensionality of the optimization problem, but crucially also reduces the time complexity of posterior inference and model selection to *linear* in the number of training data points.

## 4.1 Algorithms and Computational Complexity

We give algorithms both for iteratively constructed dense actions (Algorithm S1), as used in CaGP-CG, and for sparse batch actions (Algorithm S2), as used for CaGP-Opt, in the supplementary

material.[5] The time complexity is $\mathcal{O}(n^2 i)$ for dense actions and $\mathcal{O}(ni \max(i,k))$ for sparse actions, where $k$ is the maximum number of non-zero entries per column of $\boldsymbol{S}_i$. Both have the same linear memory requirement: $\mathcal{O}(ni)$. Since the training loss $\ell_{\text{CaGP}}^{\text{ELBO}}$ only involves terms that are also present in the posterior predictive, both model selection and predictions incur the same complexity.

## 4.2 Related Work

**Computational Uncertainty and Probabilistic Numerics**   All CaGP variants discussed in this paper fall into the category of probabilistic numerical methods [25–27], which aim to quantify approximation error arising from limited computational resources via additional uncertainty about the quantity of interest [e.g. 28–31]. Specifically, the iterative formulation of CaGP (i.e. Algorithm S1) originally proposed by Wenger et al. [19] employs a probabilistic linear solver [32–35].

**Scalable GPs with Lower-Bounded Log Marginal Likelihoods**   Numerous scalable GP approximations beyond those in Sections 1 and 2 exist; see Liu et al. [36] for a comprehensive review. Many GP models [e.g., 4, 5, 37–39] learn hyperparameters through maximizing variational lower bounds in the same spirit as SGPR, SVGP and our method. Similar to our work, interdomain inducing point methods [40–42] learn a variational posterior approximation on a small set of linear functionals applied to the latent GP. However, unlike our method, their resulting approximate posterior is usually prone to underestimating uncertainty in the same manner as SGPR and SVGP. Finally, similar to our proposed training loss for CaGP-CG, Artemev et al. [43] demonstrate how one can use the method of conjugate gradients to obtain a tighter lower bound on the log marginal likelihood.

**GP Approximation Biases and Computational Uncertainty**   Scalable GP methods inevitably introduce approximation error and thus yield biased hyperparameters and predictive distributions, with an exception of Potapczynski et al. [12] which trade bias for increased variance. Numerous works have studied pathologies associated with optimizing variational lower bounds, especially in the context of SVGP [12, 16–18], and various remedies have been proposed. In order to mitigate biases from approximation, several works alternatively propose replacing variational lower bounds with alternative model selection objectives, including leave-one-out cross-validation [44] and losses that directly target predictive performance [45, 46].

## 5   Experiments

We benchmark the generalization of computation-aware GPs with two different action choices, CaGP-Opt (ours) and CaGP-CG [19], using our proposed training objective in Equation (10) on a range of UCI datasets for regression [53]. We compare against SVGP [5], often considered to be state-of-the-art for large-scale GP regression. Per recommendations by Ober et al. [15], we also include SGPR [4] as a strong baseline for all datasets where this is computationally feasible. We also train Conjugate Gradient-based GPs (CGGP) [e.g. 7, 9, 10] using the training procedure proposed by Wenger et al. [10]. Note that CaGP-CG recovers CGGP in its posterior mean and produces nearly identical predictive error at half the computational cost for inference [Sec. 2.1 & 4 of 19], which is why the main difference between CaGP-CG and CGGP in our experiments is the training objective. Finally, we also train an exact CholeskyGP on the smallest datasets, where this is still feasible.

**Experimental Details**   All datasets were randomly partitioned into train and test sets using a $(0.9, 0.1)$ split for five random seeds. We used a zero-mean GP prior and a Matérn($3/2$) kernel with an outputscale $o^2$ and one lengthscale per input dimension $l_j^2$, as well as a scalar observation noise for the likelihood $\sigma^2$, s.t. $\boldsymbol{\theta} = (o, l_1, \ldots, l_d, \sigma) \in \mathbb{R}^{d+2}$. We used the existing implementations of SGPR, SVGP and CGGP in GPyTorch [7] and also implemented CaGP in this framework (see Section S4.2 for our open-source implementation). For SGPR and SVGP we used $m = 1024$ inducing points and for CGGP, CaGP-CG and CaGP-Opt we chose $i = 512$. We optimized the hyperparameters $\boldsymbol{\theta}$ either with Adam [54] for a maximum of 1000 epochs in `float32` or with LBFGS [55] for 100 epochs in `float64`, depending on the method and problem scale. On the largest dataset "Power", we used 400 epochs for SVGP and 200 for CaGP-Opt due to resource constraints. For SVGP we used a batch size of 1024 throughout. We scheduled the learning rate via PyTorch's [56] `LinearLR(end_factor=0.1)` scheduler for all methods and performed a hyperparameter sweep

---

[5]For a detailed analysis see Algorithms S1 and S2 in the supplementary material, which contain line-by-line time complexity and memory analyses.

Table 1: *Generalization error (NLL, RMSE, and wall-clock time) on UCI benchmark datasets.* The table shows the best results for all methods across learning rate sweeps, averaged across five random seeds. We report the epoch where each method obtained the lowest average test NLL, and all performance metrics (NLL, RMSE, and wall-clock runtime) are reported for this epoch. Highlighted in bold and color are the best *approximate* methods per metric (difference $> 1$ standard deviation).

| Dataset | $n$ | $d$ | Method | Optim. | LR | Epoch | Test NLL ↓ mean | std | Test RMSE ↓ mean | std | Avg. Runtime ↓ |
|---|---|---|---|---|---|---|---|---|---|---|---|
| Parkinsons [47] | 5 288 | 21 | CholeskyGP | LBFGS | 0.100 | 100 | -3.645 | 0.002 | 0.001 | 0.000 | 1min 3s |
| | | | SGPR | Adam | 0.100 | 268 | -2.837 | 0.087 | 0.031 | 0.022 | 27s |
| | | | | LBFGS | 1.000 | 100 | -3.245 | 0.067 | 0.007 | 0.003 | 2min 14s |
| | | | SVGP | Adam | 0.100 | 1000 | -2.858 | 0.016 | 0.006 | 0.002 | 2min 25s |
| | | | CGGP | LBFGS | 0.100 | 81 | -2.663 | 0.141 | 0.019 | 0.013 | 1min 12s |
| | | | CaGP-CG | Adam | 1.000 | 250 | -2.936 | 0.007 | 0.009 | 0.006 | 1min 44s |
| | | | CaGP-Opt | Adam | 1.000 | 956 | -3.384 | 0.005 | 0.004 | 0.002 | 1min 27s |
| | | | | LBFGS | 0.010 | 37 | **-3.449** | 0.009 | **0.002** | 0.000 | 1min 53s |
| Bike [48] | 15 642 | 16 | CholeskyGP | LBFGS | 0.100 | 100 | -3.472 | 0.012 | 0.006 | 0.007 | 7min 15s |
| | | | SGPR | Adam | 0.100 | 948 | -2.121 | 0.110 | 0.026 | 0.004 | 4min 3s |
| | | | | LBFGS | 1.000 | 100 | **-3.017** | 0.022 | **0.009** | 0.002 | 4min 10s |
| | | | SVGP | Adam | 0.010 | 1000 | -2.256 | 0.020 | 0.020 | 0.002 | 6min 41s |
| | | | CGGP | LBFGS | 1.000 | 15 | -1.952 | 0.078 | 0.024 | 0.004 | 2min 6s |
| | | | CaGP-CG | Adam | 1.000 | 250 | -2.042 | 0.024 | 0.024 | 0.002 | 5min 17s |
| | | | CaGP-Opt | Adam | 1.000 | 1000 | -2.401 | 0.037 | 0.018 | 0.002 | 8min 10s |
| | | | | LBFGS | 1.000 | 100 | -2.438 | 0.038 | 0.017 | 0.001 | 14min 48s |
| Protein [49] | 41 157 | 9 | SGPR | Adam | 0.100 | 993 | 0.844 | 0.006 | 0.561 | 0.005 | 10min 25s |
| | | | | LBFGS | 0.100 | 96 | 0.846 | 0.006 | 0.562 | 0.005 | 6min 56s |
| | | | SVGP | Adam | 0.010 | 996 | 0.851 | 0.006 | 0.564 | 0.005 | 17min 19s |
| | | | CGGP | LBFGS | 0.100 | 35 | 0.853 | 0.006 | **0.517** | 0.004 | 20min 15s |
| | | | CaGP-CG | Adam | 1.000 | 27 | **0.820** | 0.006 | 0.542 | 0.004 | 1min 26s |
| | | | CaGP-Opt | Adam | 0.100 | 941 | 0.829 | 0.005 | 0.545 | 0.004 | 13min 48s |
| | | | | LBFGS | 1.000 | 84 | 0.830 | 0.005 | 0.545 | 0.004 | 14min 29s |
| KEGGu [50] | 57 248 | 26 | SGPR | Adam | 0.100 | 143 | -0.681 | 0.025 | 0.123 | 0.002 | 2min 4s |
| | | | | LBFGS | 1.000 | 100 | **-0.712** | 0.028 | **0.118** | 0.003 | 8min 58s |
| | | | SVGP | Adam | 0.010 | 988 | **-0.710** | 0.026 | **0.118** | 0.003 | 24min 21s |
| | | | CGGP | LBFGS | 0.100 | 30 | -0.512 | 0.034 | **0.120** | 0.003 | 33min 55s |
| | | | CaGP-CG | Adam | 1.000 | 229 | **-0.699** | 0.026 | **0.120** | 0.003 | 39min 5s |
| | | | CaGP-Opt | Adam | 1.000 | 990 | **-0.693** | 0.026 | **0.120** | 0.003 | 22min 3s |
| | | | | LBFGS | 0.010 | 40 | **-0.694** | 0.026 | **0.120** | 0.003 | 22min 0s |
| Road [51] | 391 387 | 2 | SVGP | Adam | 0.001 | 998 | 0.149 | 0.007 | 0.277 | 0.002 | 2h 7min 37s |
| | | | CaGP-Opt | Adam | 0.100 | 1000 | **-0.291** | 0.011 | **0.159** | 0.003 | 2h 11min 31s |
| Power [52] | 1 844 352 | 7 | SVGP | Adam | 0.010 | 399 | **-2.104** | 0.007 | **0.029** | 0.000 | 5h 7min 57s |
| | | | CaGP-Opt | Adam | 0.100 | 200 | **-2.103** | 0.006 | 0.030 | 0.000 | 4h 32min 48s |

for the (initial) learning rate. All experiments were run on an NVIDIA Tesla V100-PCIE-32GB GPU, except for "Power", where we used an NVIDIA A100 80GB PCIe GPU to have sufficient memory for CaGP-Opt with $i = 512$. Our exact experiment configuration can be found in Table S1.

**Evaluation Metrics**    We evaluate the generalization performance once per epoch on the test set by computing the (average) negative log-likelihood (NLL) and the root mean squared error (RMSE), as well as recording the wallclock runtime. Runtime is measured at the epoch with the best average performance across random seeds.

**CaGP-Opt Matches or Outperforms SVGP**    Table 1 and Figure 3 show generalization performance of all methods for the best choice of learning rate. In terms of both NLL and RMSE, CaGP-Opt outperforms or matches the variational baselines SGPR and SVGP at comparable runtime (except on "Bike"). SGPR remains competitive for smaller datasets; however, it does not scale to the largest datasets. There are some datasets and metrics in which specific methods dominate, for example on "Bike" SGPR outperforms all other approximations, while on "Protein" methods based on CG, i.e. CGGP and CaGP-CG, perform the best. However, CaGP-Opt consistently performs either best or second best and scales to over a million datapoints. These results are quite remarkable for numerous reasons. First, CaGP is comparable in runtime to SVGP on large datasets despite the fact that it incurs a linear-time computational complexity while SVGP is constant time.[6] Second, while all of the methods we compare approximate the GP posterior with low-rank updates, CaGP-Opt (with

---

[6]While SVGP is arguably linear time since it will eventually loop through all training data, each computation of the ELBO uses a constant time stochastic approximation.

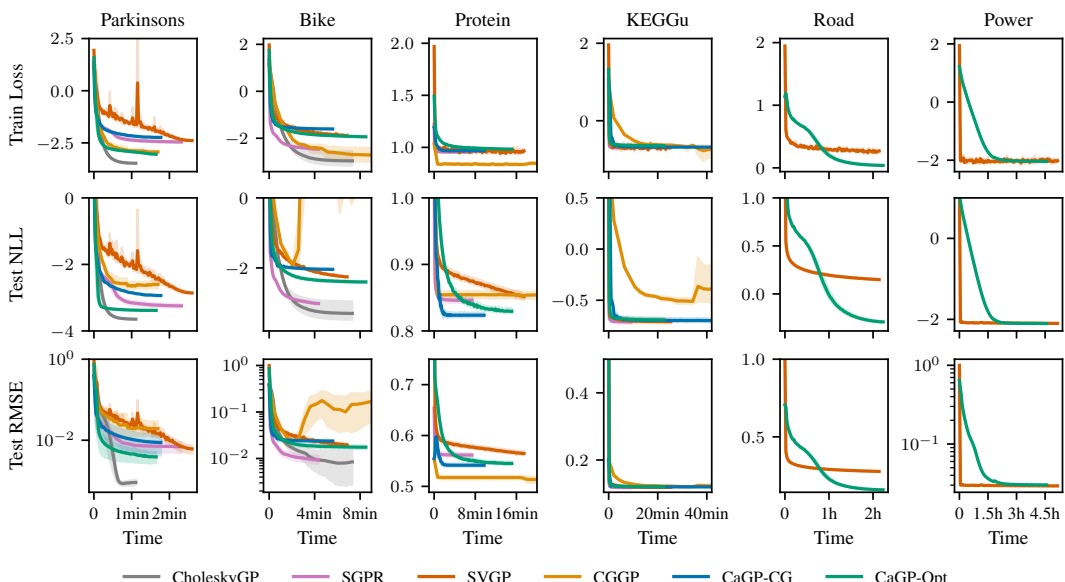

Figure 3: *Learning curves of GP approximation methods on UCI benchmark datasets.* Rows show train and test loss as a function of wall-clock time for the best choice of learning rate per method. CaGP-Opt generally displays a "ramp-up" phase early in training where performance is worse than that of SVGP. As training progresses, CaGP-Opt matches or surpasses SVGP performance.

$i = 512$) here uses half the rank of SGPR/SVGP $m = 1024$. Nevertheless, CaGP-Opt is able to substantially outperform SVGP even on spatial datasets like 3DRoad where low-rank posterior updates are often poor [57]. These results suggest that CaGP-Opt can be a more efficient approximation than inducing point methods, and that low-rank GP approximations may be more applicable than previously thought [58, 59]. Figure 3 shows the NLL and RMSE learning curves for the best choice of learning rate per method. CaGP-Opt often shows a "ramp-up" phase, compared to SVGP, but then improves or matches its generalization performance. This gap is particularly large on "Road", where CaGP-Opt is initially worse than SVGP but dominates in the second half of training.

**SVGP Overestimates Observation Noise and (Often) Lengthscale**   In Figure S5 we show that SVGP typically learns larger observation noise than other methods as suggested by previous work [18, 45] and hinted at by observations on synthetic data in Figure 1 and Figure S3(b). Additionally on larger datasets SVGP also often learns large lengthscales, which in combination with a large observation noise can lead to an oversmoothing effect. In contrast, CaGP-Opt generally learns lower observational noise than SVGP. Of course, learning a small observation noise, in particular, is important for achieving low RMSE and thus also NLL, and points to why we should expect CaGP-Opt to outperform SVGP. These hyperparameter results suggest that CaGP-Opt interprets more of the data as signal, while SVGP interprets more of the data as noise.

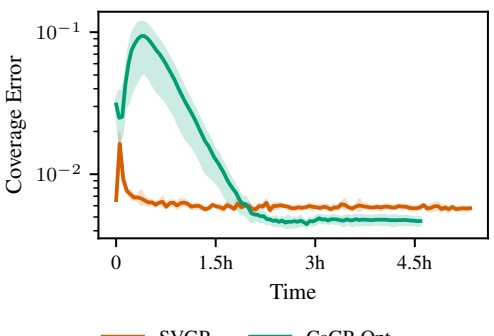

Figure 4: *Uncertainty quantification for CaGP-Opt and SVGP.* Difference between the desired coverage (95%) and the empirical coverage of the GP 95% credible interval on the "Power" dataset. After training, CaGP-Opt has better empirical coverage than SVGP.

**CaGP Improves Uncertainty Quantification Over SVGP**   A key advantage of CaGP-Opt and CaGP-CG is that their posterior uncertainty estimates capture both the uncertainty due to limited data and due to limited computation. To that end, we assess the frequentist coverage of CaGP-Opt's uncertainty estimates. We report the absolute difference between a desired coverage percentage $\alpha$

and the fraction of data that fall into the $\alpha$ credible interval of the CaGP-Opt posterior; i.e. $\varepsilon_{\text{coverage}}^{\alpha} = |\alpha - \frac{1}{n_{\text{test}}} \sum_{i=1}^{n_{\text{test}}} 1(y_i \in I_{q(\boldsymbol{x}_i)}^{\alpha})|$. Figure 4 compares the 95% coverage error for both CaGP-Opt and SVGP on the largest dataset ("Power"). From this plot, we see that the CaGP credible intervals are more aligned with the desired coverage. We hypothesize that these results reflect the different uncertainty properties of the methods: CaGP-Opt overestimates posterior uncertainty while SVGP is prone towards overconfidence.

## 6    Conclusion

In this work, we introduced strategies for model selection and posterior inference for computation-aware Gaussian processes, which scale linearly with the number of training data rather than quadratically. The key technical innovations being a sparse projection of the data, which balances minimizing posterior entropy and computational cost, and a scalable way to optimize kernel hyperparameters, both of which are amenable to GPU acceleration. All together, these advances enable competitive or improved performance over previous approximate inference methods on large-scale datasets, in terms of generalization and uncertainty quantification. Remarkably, our method outperforms SVGP—often considered the de-facto GP approximation standard— even when compressing the data into a space of half the dimension of the variational parameters. Finally, we also demonstrate that computation-aware GPs avoid many of the pathologies often observed in inducing point methods, such as overconfidence and oversmoothing.

**Limitations**    While CaGP-Opt obtains the same linear time and memory costs as SGPR, it is not amenable to stochastic minibatching and thus cannot achieve the constant time/memory capabilities of SVGP. In practice, this asymptotic difference does not result in substantially different wall clock times, as SVGP requires many more optimizer steps than CaGP-Opt due to batching. (Indeed, on many datasets we find that CaGP-Opt is faster.) CaGP-Opt nevertheless requires larger GPUs than SVGP on datasets with more than a million data points. Moreover, tuning CaGP-Opt requires choosing the appropriate number of actions (i.e. the rank of the approximate posterior update), though we note that most scalable GP approximations have a similar tunable parameter (e.g. number of inducing points). Perhaps the most obvious limitation is that CaGP, unlike SVGP, is limited to GP regression with a conjugate observational noise model. We leave extensions to classification and other non-conjugate likelihoods as future work.

**Outlook and Future Work**    An immediate consequence of this work is the ability to apply computation-aware Gaussian processes to "real-world" problems, as our approach solves CaGP's open problems of model selection and scalability. Looking forward, an exciting future vision is a general framework for problems involving a Gaussian process model with a downstream task where the actions are chosen optimally, given resource constraints, to solve said task. Future work will pursue this direction beyond Gaussian likelihoods to non-conjugate models and downstream tasks such as Bayesian optimization.

## Acknowledgments and Disclosure of Funding

JW and JPC are supported by the Gatsby Charitable Foundation (GAT3708), the Simons Foundation (542963), the NSF AI Institute for Artificial and Natural Intelligence (ARNI: NSF DBI 2229929) and the Kavli Foundation. JG and KW are supported by the NSF (IIS-2145644, DBI-2400135). PH gratefully acknowledges co-funding by the European Union (ERC, ANUBIS, 101123955). Views and opinions expressed are however those of the author(s) only and do not necessarily reflect those of the European Union or the European Research Council. Neither the European Union nor the granting authority can be held responsible for them. PH is a member of the Machine Learning Cluster of Excellence, funded by the Deutsche Forschungsgemeinschaft (DFG, German Research Foundation) under Germany's Excellence Strategy – EXC number 2064/1 – Project number 390727645; he also gratefully acknowledges the German Federal Ministry of Education and Research (BMBF) through the Tübingen AI Center (FKZ: 01IS18039A); and funds from the Ministry of Science, Research and Arts of the State of Baden-Württemberg. GP acknowledges support from NSERC and the Canada CIFAR AI Chair program.

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

# Supplementary Material

This supplementary material contains additional results and in particular proofs for all theoretical statements. References referring to sections, equations or theorem-type environments within this document are prefixed with 'S', while references to, or results from, the main paper are stated as is.

# S1  Theoretical Results

## S1.1  Alternative Derivation of CaGP Posterior

**Lemma S1** (CaGP Inference as Exact Inference Given a Modified Observation Model)
*Given a Gaussian process prior $f \sim \mathcal{GP}(\mu, K)$ and training data $(\boldsymbol{X}, \boldsymbol{y})$ the computation-aware GP posterior $\mathcal{GP}(\mu_i, K_i)$ (see Equation (5)) with linearly independent and fixed actions $\boldsymbol{S}$ is equivalent to an exact batch GP posterior $(f \mid \tilde{\boldsymbol{y}})$ given data $\tilde{\boldsymbol{y}} = \boldsymbol{S'}^\mathsf{T} \boldsymbol{y}$ observed according to the likelihood $\tilde{\boldsymbol{y}} \mid f(\boldsymbol{X}) \sim \mathcal{N}\left(\boldsymbol{S'}^\mathsf{T} f(\boldsymbol{X}), \sigma^2 \boldsymbol{I}\right)$, where $\boldsymbol{S'} = \boldsymbol{S} \operatorname{chol}(\boldsymbol{S}^\mathsf{T} \boldsymbol{S})^{-\mathsf{T}}$.*

*Proof.* First note that by definition $\boldsymbol{S'}$ has orthonormal columns, since

$$
\begin{aligned}
\boldsymbol{S'}^\mathsf{T} \boldsymbol{S'} &= (\boldsymbol{S} \operatorname{chol}(\boldsymbol{S}^\mathsf{T} \boldsymbol{S})^{-\mathsf{T}})^\mathsf{T} \boldsymbol{S} \operatorname{chol}(\boldsymbol{S}^\mathsf{T} \boldsymbol{S})^{-\mathsf{T}} \\
&= \operatorname{chol}(\boldsymbol{S}^\mathsf{T} \boldsymbol{S})^{-1} \boldsymbol{S}^\mathsf{T} \boldsymbol{S} \operatorname{chol}(\boldsymbol{S}^\mathsf{T} \boldsymbol{S})^{-\mathsf{T}} \\
&= \boldsymbol{L}^{-1} \boldsymbol{L} \boldsymbol{L}^\mathsf{T} \boldsymbol{L}^{-\mathsf{T}} \\
&= \boldsymbol{I}.
\end{aligned}
$$

Now by basic properties of Gaussian distributions, we have for arbitrary $\boldsymbol{X}_\diamond \in \mathbb{R}^{n_\diamond \times d}$ that

$$
\begin{pmatrix} \tilde{\boldsymbol{y}} \\ f(\boldsymbol{X}_\diamond) \end{pmatrix} \sim \mathcal{N}\left( \begin{pmatrix} \boldsymbol{S'}^\mathsf{T} \mu(\boldsymbol{X}) \\ \mu(\boldsymbol{X}_\diamond) \end{pmatrix}, \begin{pmatrix} \boldsymbol{S'}^\mathsf{T} K(\boldsymbol{X}, \boldsymbol{X}) \boldsymbol{S'} + \sigma^2 \boldsymbol{S'}^\mathsf{T} \boldsymbol{S'} & \boldsymbol{S'}^\mathsf{T} K(\boldsymbol{X}, \boldsymbol{X}_\diamond) \\ K(\boldsymbol{X}_\diamond, \boldsymbol{X}) \boldsymbol{S'} & K(\boldsymbol{X}_\diamond, \boldsymbol{X}_\diamond) \end{pmatrix} \right)
$$

is jointly Gaussian, where we used that $\boldsymbol{I} = \boldsymbol{S'}^\mathsf{T} \boldsymbol{S'}$.

Therefore we have that $(f(\boldsymbol{X}_\diamond) \mid \tilde{\boldsymbol{y}}) \sim \mathcal{N}(\mu_i(\boldsymbol{X}_\diamond), K_i(\boldsymbol{X}_\diamond, \boldsymbol{X}_\diamond))$ with

$$
\begin{aligned}
\mu_i(\boldsymbol{X}_\diamond) &= \mu(\boldsymbol{X}_\diamond) + K(\boldsymbol{X}_\diamond, \boldsymbol{X}) \boldsymbol{S'} (\boldsymbol{S'}^\mathsf{T} (K(\boldsymbol{X}, \boldsymbol{X}) + \sigma^2 \boldsymbol{I}) \boldsymbol{S'})^{-1} (\tilde{\boldsymbol{y}} - \boldsymbol{S'}^\mathsf{T} \boldsymbol{\mu}), \\
&= \mu(\boldsymbol{X}_\diamond) + K(\boldsymbol{X}_\diamond, \boldsymbol{X}) \boldsymbol{S} (\boldsymbol{S}^\mathsf{T} (K(\boldsymbol{X}, \boldsymbol{X}) + \sigma^2 \boldsymbol{I}) \boldsymbol{S})^{-1} \boldsymbol{S}^\mathsf{T} (\boldsymbol{y} - \boldsymbol{\mu})
\end{aligned}
$$

$$K_i(\boldsymbol{X}_\diamond, \boldsymbol{X}_\diamond) = K(\boldsymbol{X}_\diamond, \boldsymbol{X}_\diamond) - K(\boldsymbol{X}_\diamond, \boldsymbol{X})\boldsymbol{S}'(\boldsymbol{S}'^\mathsf{T}(K(\boldsymbol{X}, \boldsymbol{X}) + \sigma^2\boldsymbol{I})\boldsymbol{S}')^{-1}\boldsymbol{S}'^\mathsf{T}K(\boldsymbol{X}, \boldsymbol{X}_\diamond)$$
$$= K(\boldsymbol{X}_\diamond, \boldsymbol{X}_\diamond) - K(\boldsymbol{X}_\diamond, \boldsymbol{X})\boldsymbol{S}(\boldsymbol{S}^\mathsf{T}(K(\boldsymbol{X}, \boldsymbol{X}) + \sigma^2\boldsymbol{I})\boldsymbol{S})^{-1}\boldsymbol{S}^\mathsf{T}K(\boldsymbol{X}, \boldsymbol{X}_\diamond)$$

which is equivalent to the definition of the CaGP posterior in Equation (5). This proves the claim.

$\square$

## S1.2 Worst Case Error Interpretations of the Variance of Exact GPs, CaGPs and SVGPs

In order to understand the impact of approximation on the uncertainty quantification of both CaGP and SVGP, it is instructive to compare the theoretical guarantees they admit, when the latent function is assumed to be in the RKHS of the kernel. In the context of model selection, this corresponds to the ideal case where the optimization has converged to the ground truth hyperparameters.

Let $f \sim \mathcal{GP}(0, K)$ be a Gaussian process with kernel $K$ and define the observed process $y(\cdot) = f(\cdot) + \sigma\varepsilon(\cdot)$ where $\varepsilon \sim \mathcal{GP}(0, \delta)$ is white noise with noise level $\sigma^2 > 0$, i.e. $\delta(\boldsymbol{x}, \boldsymbol{x}') = 1(\boldsymbol{x} = \boldsymbol{x}')$. Consequently, the covariance kernel of the data-generating process $y(\cdot)$ is given by $K^\sigma(\boldsymbol{x}, \boldsymbol{x}') := K(\boldsymbol{x}, \boldsymbol{x}') + \sigma^2\delta(\boldsymbol{x}, \boldsymbol{x}')$ and we denote the corresponding RKHS as $\mathbb{H}_{K^\sigma}$.

For exact Gaussian process inference, the pointwise (relative) worst-case squared error of the posterior mean is precisely given by the posterior predictive variance.

**Theorem S1** (Worst Case Error Interpretation of GP Variance [60])
*Given a set of training inputs $\boldsymbol{x}_1, \ldots, \boldsymbol{x}_n \in \mathbb{X}$, the GP posterior $\mathcal{GP}(\mu_\star, K_\star)$ satisfies for any $\boldsymbol{x} \neq \boldsymbol{x}_j$ that*

$$\sup_{y \in \mathbb{H}_{K^\sigma}, \|y\|_{\mathbb{H}_{K^\sigma}} \leq 1} \underbrace{(y(\boldsymbol{x}_\diamond) - \mu_\star^y(\boldsymbol{x}_\diamond))^2}_{\text{error of posterior mean}} = \underbrace{K_\star(\boldsymbol{x}_\diamond, \boldsymbol{x}_\diamond) + \sigma^2}_{\text{predictive variance}} \tag{S15}$$

*If $\sigma^2 = 0$, then the above also holds for $\boldsymbol{x}_\diamond = \boldsymbol{x}_j$.*

*Proof.* See Proposition 3.8 of Kanagawa et al. [60]. $\square$

CaGP admits precisely the same guarantee just with the *approximate* posterior mean and covariance function. The fact that the impact of the approximation on the posterior mean is exactly captured by the approximate predictive variance function is what is meant by the method being *computation-aware*.

**Theorem S2** (Worst Case Error Interpretation of CaGP Variance [19])
*Given a set of training inputs $\boldsymbol{x}_1, \ldots, \boldsymbol{x}_n \in \mathbb{X}$, the CaGP posterior $\mathcal{GP}(\mu_i, K_i)$ satisfies for any $\boldsymbol{x} \neq \boldsymbol{x}_j$ that*

$$\sup_{y \in \mathbb{H}_{K^\sigma}, \|y\|_{\mathbb{H}_{K^\sigma}} \leq 1} \underbrace{(y(\boldsymbol{x}_\diamond) - \mu_i^y(\boldsymbol{x}_\diamond))^2}_{\text{error of approximate posterior mean}} = \underbrace{K_i(\boldsymbol{x}_\diamond, \boldsymbol{x}_\diamond) + \sigma^2}_{\text{approximate predictive variance}} \tag{S16}$$

*If $\sigma^2 = 0$, then the above also holds for $\boldsymbol{x}_\diamond = \boldsymbol{x}_j$.*

*Proof.* See Theorem 2 of Wenger et al. [19]. $\square$

While SVGP also admits a decomposition of its approximate predictive variance into two (relative) worst-case errors, neither of these is the error we care about, namely the difference between the data-generating function $y \in \mathbb{H}_{K^\sigma}$ and the approximate posterior mean $\mu_{\text{SVGP}}(\boldsymbol{x}_\diamond)$. It only involves a worst-case error term over the unit ball in the RKHS of the *approximate* kernel $Q^\sigma \approx K^\sigma$.

**Theorem S3** (Worst Case Error Interpretation of SVGP Variance [61])
*Given a set of training inputs $\boldsymbol{x}_1, \ldots, \boldsymbol{x}_n \in \mathbb{X}$ and (fixed) inducing points $\boldsymbol{Z} \in \mathbb{R}^{m \times d}$, the optimal variational posterior $\mathcal{GP}(\mu_{\text{SVGP}}^\star, K_{\text{SVGP}}^\star)$ of SVGP is given by*

$$\mu_{\text{SVGP}}^{\star, y}(\boldsymbol{x}_\diamond) = K(\boldsymbol{x}_\diamond, \boldsymbol{Z})(\sigma^2 K(\boldsymbol{Z}, \boldsymbol{Z}) + K(\boldsymbol{Z}, \boldsymbol{X})K(\boldsymbol{X}, \boldsymbol{Z}))^{-1}K(\boldsymbol{Z}, \boldsymbol{X})y(\boldsymbol{X})$$
$$K_{\text{SVGP}}^\star(\boldsymbol{x}_\diamond, \boldsymbol{x}_\diamond') = K(\boldsymbol{x}_\diamond, \boldsymbol{x}_\diamond') - Q(\boldsymbol{x}_\diamond, \boldsymbol{x}_\diamond') + K(\boldsymbol{x}_\diamond, \boldsymbol{Z})(K(\boldsymbol{Z}, \boldsymbol{Z}) + \sigma^{-2}K(\boldsymbol{Z}, \boldsymbol{X})K(\boldsymbol{X}, \boldsymbol{Z}))^{-1}K(\boldsymbol{Z}, \boldsymbol{x}_\diamond')$$

*where $Q(\boldsymbol{x}, \boldsymbol{x}') = K(\boldsymbol{x}, \boldsymbol{Z})K(\boldsymbol{Z}, \boldsymbol{Z})^{-1}K(\boldsymbol{Z}, \boldsymbol{x}')$ is the Nyström approximation of the covariance function $K(\boldsymbol{x}, \boldsymbol{x}')$ (see Eqns. (25) and (26) of Wild et al. [61]). The optimized SVGP posterior*

*satisfies for any $\boldsymbol{x} \neq \boldsymbol{x}_j$ that*

$$
\sup_{y \in \mathbb{H}_{Q^\sigma}, \|h\|_{\mathbb{H}_{Q^\sigma}} \leq 1} \underbrace{\left(y(\boldsymbol{x}_\diamond) - \mu^{\star,y}_{\text{SVGP}}(\boldsymbol{x}_\diamond)\right)^2}_{\text{error of exact posterior mean \color{red} assuming } y(\cdot) \text{ \color{red} is in the RKHS of the approximate kernel } Q^\sigma}
$$

$$
+ \sup_{f \in \mathbb{H}_K, \|f\|_{\mathbb{H}_K} \leq 1} \underbrace{\left(f(\boldsymbol{x}_\diamond) - K(\boldsymbol{x}_\diamond, \boldsymbol{Z})K(\boldsymbol{Z}, \boldsymbol{Z})^{-1}f(\boldsymbol{Z})\right)^2}_{\text{error of exact posterior mean given noise-free observations at inducing points}} \qquad \text{(S17)}
$$

$$
= \underbrace{K^\star_{\text{SVGP}}(\boldsymbol{x}_\diamond, \boldsymbol{x}_\diamond) + \sigma^2}_{\text{approximate predictive variance}}
$$

*If $\sigma^2 = 0$, then the above also holds for $\boldsymbol{x}_\diamond = \boldsymbol{x}_j$.*

*Proof.* See Theorem 6 of Wild et al. [61]. □

### S1.3 CaGP's Variance Decreases Monotonically as the Number of Iterations Increases

**Proposition S1** (CaGP's Variance Decreases Monotonically with the Number of Iterations)
*Given a training dataset of size $n$, let $\mathcal{GP}(\mu_i, K_i)$ be the corresponding CaGP posterior defined in Equation* (5) *where $i \leq n$ denotes the downdate rank / number of iterations and assume the CaGP actions $\boldsymbol{S}_i \in \mathbb{R}^{n \times i}$ are linearly independent. Then it holds for arbitrary $\boldsymbol{x}_\diamond \in \mathbb{X}$ and $i \leq j \leq n$, that*

$$
K_i(\boldsymbol{x}_\diamond, \boldsymbol{x}_\diamond) \geq K_j(\boldsymbol{x}_\diamond, \boldsymbol{x}_\diamond) \geq K_n(\boldsymbol{x}_\diamond, \boldsymbol{x}_\diamond) = K_\star(\boldsymbol{x}_\diamond, \boldsymbol{x}_\diamond) \qquad \text{(S18)}
$$

*where $K_\star(\boldsymbol{x}_\diamond, \boldsymbol{x}_\diamond)$ is the variance of the exact GP posterior in Equation* (1).

*Proof.* Wenger et al. [19] originally defined the approximate precision matrix $\boldsymbol{C}_i = \sum_{\ell=1}^i \frac{1}{\eta_\ell}\boldsymbol{d}_\ell\boldsymbol{d}_\ell^\mathsf{T} = \sum_{\ell=1}^i \tilde{\boldsymbol{d}}_\ell\tilde{\boldsymbol{d}}_\ell^\mathsf{T}$ as a sum of rank-1 matrices and show that this definition is equivalent to the batch form $\boldsymbol{C}_i = \boldsymbol{S}_i(\boldsymbol{S}_i^\mathsf{T}\hat{\boldsymbol{K}}\boldsymbol{S}_i)^{-1}\boldsymbol{S}_i^\mathsf{T}$ we use in this work [see Lemma S1, Eqn. (S37) in 19]. Therefore we have that

$$
K_i(\boldsymbol{x}_\diamond, \boldsymbol{x}_\diamond) = K(\boldsymbol{x}_\diamond, \boldsymbol{x}_\diamond) - K(\boldsymbol{x}_\diamond, \boldsymbol{X})\boldsymbol{C}_i K(\boldsymbol{X}, \boldsymbol{x}_\diamond)
$$

$$
= K(\boldsymbol{x}_\diamond, \boldsymbol{x}_\diamond) - \sum_{\ell=1}^i K(\boldsymbol{x}_\diamond, \boldsymbol{X})\tilde{\boldsymbol{d}}_\ell\tilde{\boldsymbol{d}}_\ell^\mathsf{T} K(\boldsymbol{X}, \boldsymbol{x}_\diamond)
$$

$$
= K(\boldsymbol{x}_\diamond, \boldsymbol{x}_\diamond) - \sum_{\ell=1}^i (K(\boldsymbol{x}_\diamond, \boldsymbol{X})\tilde{\boldsymbol{d}}_\ell)^2
$$

$$
\geq K(\boldsymbol{x}_\diamond, \boldsymbol{x}_\diamond) - \sum_{\ell=1}^j (K(\boldsymbol{x}_\diamond, \boldsymbol{X})\tilde{\boldsymbol{d}}_\ell)^2 \qquad \text{since } i \leq j
$$

$$
\geq K(\boldsymbol{x}_\diamond, \boldsymbol{x}_\diamond) - \sum_{\ell=1}^n (K(\boldsymbol{x}_\diamond, \boldsymbol{X})\tilde{\boldsymbol{d}}_\ell)^2
$$

$$
= K(\boldsymbol{x}_\diamond, \boldsymbol{x}_\diamond) - K(\boldsymbol{x}_\diamond, \boldsymbol{X})\boldsymbol{C}_n K(\boldsymbol{X}, \boldsymbol{x}_\diamond)
$$

$$
= K_\star(\boldsymbol{x}_\diamond, \boldsymbol{x}_\diamond)
$$

where the last equality follows from the fact that $\boldsymbol{S}_n \in \mathbb{R}^{n \times n}$ has rank $n$ and therefore

$$
\boldsymbol{C}_n = \boldsymbol{S}_n(\boldsymbol{S}_n^\mathsf{T}\hat{\boldsymbol{K}}\boldsymbol{S}_n)^{-1}\boldsymbol{S}_n^\mathsf{T} = \boldsymbol{S}_n\boldsymbol{S}_n^{-1}\hat{\boldsymbol{K}}^{-1}(\boldsymbol{S}_n\boldsymbol{S}_n^{-1})^\mathsf{T} = \hat{\boldsymbol{K}}^{-1}.
$$

□

## S2 Training Losses

### S2.1 Projected-Data Log-Marginal Likelihood

**Lemma S2** (Projected-Data Log-Marginal Likelihood)
*Under the assumptions of Lemma S1, the* projected-data log-marginal likelihood *is given by*

$$
\ell^{\text{NLL}}_{\text{proj}}(\boldsymbol{\theta}) = -\log p(\tilde{\boldsymbol{y}} \mid \boldsymbol{\theta}) = -\log \mathcal{N}\left(\tilde{\boldsymbol{y}}; {\boldsymbol{S}'_i}^\mathsf{T}\boldsymbol{\mu}, {\boldsymbol{S}'_i}^\mathsf{T}\hat{\boldsymbol{K}}\boldsymbol{S}'_i\right)
$$

$$= \frac{1}{2}\big((\boldsymbol{y}-\boldsymbol{\mu})^\mathsf{T}\boldsymbol{S}_i(\boldsymbol{S}_i^\mathsf{T}\hat{\boldsymbol{K}}\boldsymbol{S}_i)^{-1}\boldsymbol{S}_i^\mathsf{T}(\boldsymbol{y}-\boldsymbol{\mu}) + \log\det(\boldsymbol{S}_i^\mathsf{T}\hat{\boldsymbol{K}}\boldsymbol{S}_i) - \log\det(\boldsymbol{S}_i^\mathsf{T}\boldsymbol{S}_i) + i\log(2\pi)\big)$$

*Proof.* By the same argument as in Lemma S1 we obtain that

$$\ell_{\mathrm{proj}}^{\mathrm{NLL}}(\boldsymbol{\theta}) = -\log p(\tilde{\boldsymbol{y}} \mid \boldsymbol{\theta}) = -\log\mathcal{N}\Big(\tilde{\boldsymbol{y}}; {\boldsymbol{S}_i'}^\mathsf{T}\boldsymbol{\mu}, {\boldsymbol{S}_i'}^\mathsf{T}\hat{\boldsymbol{K}}\boldsymbol{S}_i'\Big)$$

$$= \frac{1}{2}\big(({\boldsymbol{S}_i'}^\mathsf{T}\boldsymbol{y} - {\boldsymbol{S}_i'}^\mathsf{T}\boldsymbol{\mu})^\mathsf{T}({\boldsymbol{S}_i'}^\mathsf{T}\hat{\boldsymbol{K}}\boldsymbol{S}_i')^{-1}({\boldsymbol{S}_i'}^\mathsf{T}\boldsymbol{y} - {\boldsymbol{S}_i'}^\mathsf{T}\boldsymbol{\mu}) + \log\det({\boldsymbol{S}_i'}^\mathsf{T}\hat{\boldsymbol{K}}\boldsymbol{S}_i') + i\log(2\pi)\big)$$

$$= \frac{1}{2}\big((\boldsymbol{y}-\boldsymbol{\mu})^\mathsf{T}\boldsymbol{S}_i'({\boldsymbol{S}_i'}^\mathsf{T}\hat{\boldsymbol{K}}\boldsymbol{S}_i')^{-1}{\boldsymbol{S}_i'}^\mathsf{T}(\boldsymbol{y}-\boldsymbol{\mu}) + \log\det({\boldsymbol{S}_i'}^\mathsf{T}\hat{\boldsymbol{K}}\boldsymbol{S}_i') + i\log(2\pi)\big)$$

Since $\boldsymbol{S}_i' = \boldsymbol{S}_i\boldsymbol{L}^{-\mathsf{T}}$, where $\boldsymbol{L}^{-\mathsf{T}} = \mathrm{chol}(\boldsymbol{S}_i^\mathsf{T}\boldsymbol{S}_i)^{-\mathsf{T}}$ is the orthonormalizing matrix, $\boldsymbol{L}^{-\mathsf{T}}$ cancels in the quadratic loss term, giving

$$= \frac{1}{2}\big((\boldsymbol{y}-\boldsymbol{\mu})^\mathsf{T}\boldsymbol{S}_i(\boldsymbol{S}_i^\mathsf{T}\hat{\boldsymbol{K}}\boldsymbol{S}_i)^{-1}\boldsymbol{S}_i^\mathsf{T}(\boldsymbol{y}-\boldsymbol{\mu}) + \log\det({\boldsymbol{S}_i'}^\mathsf{T}\hat{\boldsymbol{K}}\boldsymbol{S}_i') + i\log(2\pi)\big)$$

and finally we can decompose the log-determinant into a difference of log-determinants

$$= \frac{1}{2}\big((\boldsymbol{y}-\boldsymbol{\mu})^\mathsf{T}\boldsymbol{S}_i(\boldsymbol{S}_i^\mathsf{T}\hat{\boldsymbol{K}}\boldsymbol{S}_i)^{-1}\boldsymbol{S}_i^\mathsf{T}(\boldsymbol{y}-\boldsymbol{\mu}) + \log\det(\boldsymbol{S}_i^\mathsf{T}\hat{\boldsymbol{K}}\boldsymbol{S}_i) - 2\log\det(\boldsymbol{L}) + i\log(2\pi)\big)$$

$$= \frac{1}{2}\big((\boldsymbol{y}-\boldsymbol{\mu})^\mathsf{T}\boldsymbol{S}_i(\boldsymbol{S}_i^\mathsf{T}\hat{\boldsymbol{K}}\boldsymbol{S}_i)^{-1}\boldsymbol{S}_i^\mathsf{T}(\boldsymbol{y}-\boldsymbol{\mu}) + \log\det(\boldsymbol{S}_i^\mathsf{T}\hat{\boldsymbol{K}}\boldsymbol{S}_i) - \log\det(\boldsymbol{L}\boldsymbol{L}^\mathsf{T}) + i\log(2\pi)\big)$$

which using $\boldsymbol{L}\boldsymbol{L}^\mathsf{T} = \boldsymbol{S}_i^\mathsf{T}\boldsymbol{S}_i$ completes the proof. $\qquad\square$

## S2.2 Evidence Lower Bound (ELBO)

**Lemma S3** (Evidence Lower Bound Training Loss)
*Define the variational family*

$$\mathcal{Q} := \big\{q(\boldsymbol{f}) = \mathcal{N}(\boldsymbol{f}; \mu_i(\boldsymbol{X}), K_i(\boldsymbol{X}, \boldsymbol{X})) \mid \boldsymbol{S} \in \mathbb{R}^{n\times i}\big\} \tag{S19}$$

*then the* evidence lower bound (ELBO) *is given by*

$$\ell_{\mathrm{CaGP}}^{\mathrm{ELBO}}(\boldsymbol{\theta}) = -\log p(\boldsymbol{y} \mid \boldsymbol{\theta}) + \mathrm{KL}(q(\boldsymbol{f}) \parallel p(\boldsymbol{f} \mid \boldsymbol{y}))$$

$$= -\mathbb{E}_q(\log p(\boldsymbol{y} \mid \boldsymbol{f})) + \mathrm{KL}(q(\boldsymbol{f}) \parallel p(\boldsymbol{f}))$$

$$= \frac{1}{2}\bigg(\frac{1}{\sigma^2}\Big(\|\boldsymbol{y} - \mu_i(\boldsymbol{X})\|_2^2 + \sum_{j=1}^n K_i(\boldsymbol{x}_j, \boldsymbol{x}_j)\Big) + (n-i)\log(\sigma^2) + n\log(2\pi)$$

$$+ \tilde{\boldsymbol{v}}_i^\mathsf{T}\boldsymbol{S}^\mathsf{T}\boldsymbol{K}\boldsymbol{S}\tilde{\boldsymbol{v}}_i - \mathrm{tr}((\boldsymbol{S}^\mathsf{T}\hat{\boldsymbol{K}}\boldsymbol{S})^{-1}\boldsymbol{S}^\mathsf{T}\boldsymbol{K}\boldsymbol{S}) + \log\det(\boldsymbol{S}^\mathsf{T}\hat{\boldsymbol{K}}\boldsymbol{S}) - \log\det(\boldsymbol{S}^\mathsf{T}\boldsymbol{S})\bigg)$$

*where $\tilde{\boldsymbol{v}}_i = (\boldsymbol{S}^\mathsf{T}\hat{\boldsymbol{K}}\boldsymbol{S})^{-1}\boldsymbol{S}^\mathsf{T}(\boldsymbol{y}-\boldsymbol{\mu})$ are the "projected" representer weights.*

*Proof.* The ELBO is given by

$$-\ell_{\mathrm{CaGP}}^{\mathrm{ELBO}}(\boldsymbol{\theta}) = \mathbb{E}_q(\log p(\boldsymbol{y} \mid \boldsymbol{f})) - \mathrm{KL}(q(\boldsymbol{f}) \parallel p(\boldsymbol{f})).$$

We first compute the expected log-likelihood term.

$$\mathbb{E}_q(\log p(\boldsymbol{y} \mid \boldsymbol{f})) = \mathbb{E}_q\bigg(-\frac{1}{2}\Big(\frac{1}{\sigma^2}(\boldsymbol{y}-\boldsymbol{f})^\mathsf{T}(\boldsymbol{y}-\boldsymbol{f}) + \log\det(\sigma^2\boldsymbol{I}_{n\times n}) + n\log(2\pi)\Big)\bigg)$$

$$= -\frac{1}{2}\bigg(\frac{1}{\sigma^2}\mathbb{E}_q\big((\boldsymbol{y}-\boldsymbol{f})^\mathsf{T}(\boldsymbol{y}-\boldsymbol{f})\big) + n\log(\sigma^2) + n\log(2\pi)\bigg)$$

Now using $\mathbb{E}\big(\boldsymbol{x}^\mathsf{T}\boldsymbol{A}\boldsymbol{x}\big) = \mathbb{E}(\boldsymbol{x})^\mathsf{T}\boldsymbol{A}\,\mathbb{E}(\boldsymbol{x}) + \mathrm{tr}(\boldsymbol{A}\,\mathrm{Cov}(\boldsymbol{x}))$, we obtain

$$= -\frac{1}{2}\bigg(\frac{1}{\sigma^2}\Big(\|\boldsymbol{y} - \mu_i(\boldsymbol{X})\|_2^2 + \mathrm{tr}(K_i(\boldsymbol{X}, \boldsymbol{X}))\Big) + n\log(\sigma^2) + n\log(2\pi)\bigg)$$

$$= -\frac{1}{2}\left(\frac{1}{\sigma^2}\left(\|\boldsymbol{y} - \mu_i(\boldsymbol{X})\|_2^2 + \sum_{j=1}^{n} K_i(\boldsymbol{x}_j, \boldsymbol{x}_j)\right) + n\log(\sigma^2) + n\log(2\pi)\right)$$

Since both $q(\boldsymbol{f})$ and the prior $p(\boldsymbol{f})$ are Gaussian, the KL divergence term between them is given by

$$\mathrm{KL}(q(\boldsymbol{f}) \parallel p(\boldsymbol{f})) = \frac{1}{2}\left((\mu_i(\boldsymbol{X}) - \mu(\boldsymbol{X}))^\mathsf{T} \boldsymbol{K}^{-1}(\mu_i(\boldsymbol{X}) - \mu(\boldsymbol{X})) + \log\left(\frac{\det(\boldsymbol{K})}{\det(K_i(\boldsymbol{X}, \boldsymbol{X}))}\right)\right.$$

$$\left. + \operatorname{tr}(\boldsymbol{K}^{-1} K_i(\boldsymbol{X}, \boldsymbol{X})) - n\right)$$

$$= \frac{1}{2}\left((\boldsymbol{K}\boldsymbol{C}_i(\boldsymbol{y} - \mu(\boldsymbol{X})))^\mathsf{T} \boldsymbol{K}^{-1} \boldsymbol{K}\boldsymbol{C}_i(\boldsymbol{y} - \mu(\boldsymbol{X})) - \log\det(\boldsymbol{K}^{-1} K_i(\boldsymbol{X}, \boldsymbol{X}))\right.$$

$$\left. + \operatorname{tr}(\boldsymbol{I}_{n\times n} - \boldsymbol{C}_i\boldsymbol{K}) - n\right)$$

$$= \frac{1}{2}\left((\boldsymbol{y} - \mu(\boldsymbol{X}))^\mathsf{T} \boldsymbol{C}_i\boldsymbol{K}\boldsymbol{C}_i(\boldsymbol{y} - \mu(\boldsymbol{X})) - \log\det(\boldsymbol{I}_{n\times n} - \boldsymbol{C}_i\boldsymbol{K}) + \operatorname{tr}(\boldsymbol{I}_{n\times n} - \boldsymbol{C}_i\boldsymbol{K}) - n\right)$$

$$= \frac{1}{2}\left(\tilde{\boldsymbol{v}}_i^\mathsf{T} \boldsymbol{S}^\mathsf{T} \boldsymbol{K}\boldsymbol{S}\tilde{\boldsymbol{v}}_i - \log\det(\boldsymbol{I}_{n\times n} - \boldsymbol{C}_i\boldsymbol{K}) + \operatorname{tr}(\boldsymbol{I}_{n\times n} - \boldsymbol{C}_i\boldsymbol{K}) - n\right)$$

$$= \frac{1}{2}\left(\tilde{\boldsymbol{v}}_i^\mathsf{T} \boldsymbol{S}^\mathsf{T} \boldsymbol{K}\boldsymbol{S}\tilde{\boldsymbol{v}}_i - \log\det(\boldsymbol{I}_{n\times n} - \boldsymbol{C}_i\boldsymbol{K}) - \operatorname{tr}((\boldsymbol{S}^\mathsf{T}\hat{\boldsymbol{K}}\boldsymbol{S})^{-1}\boldsymbol{S}^\mathsf{T}\boldsymbol{K}\boldsymbol{S})\right)$$

Next, we use the matrix determinant lemma $\det(\boldsymbol{A} + \boldsymbol{U}\boldsymbol{V}^\mathsf{T}) = \det(\boldsymbol{I}_m + \boldsymbol{V}^\mathsf{T}\boldsymbol{A}^{-1}\boldsymbol{U})\det(\boldsymbol{A})$:

$$= \frac{1}{2}\left(\tilde{\boldsymbol{v}}_i^\mathsf{T} \boldsymbol{S}^\mathsf{T} \boldsymbol{K}\boldsymbol{S}\tilde{\boldsymbol{v}}_i - \log\det(\boldsymbol{I}_{i\times i} - (\boldsymbol{S}^\mathsf{T}\hat{\boldsymbol{K}}\boldsymbol{S})^{-1}\boldsymbol{S}^\mathsf{T}\boldsymbol{K}\boldsymbol{S}) - \operatorname{tr}((\boldsymbol{S}^\mathsf{T}\hat{\boldsymbol{K}}\boldsymbol{S})^{-1}\boldsymbol{S}^\mathsf{T}\boldsymbol{K}\boldsymbol{S})\right)$$

$$= \frac{1}{2}\left(\tilde{\boldsymbol{v}}_i^\mathsf{T} \boldsymbol{S}^\mathsf{T} \boldsymbol{K}\boldsymbol{S}\tilde{\boldsymbol{v}}_i - \log\det((\boldsymbol{S}^\mathsf{T}\hat{\boldsymbol{K}}\boldsymbol{S})^{-1}(\boldsymbol{S}^\mathsf{T}\hat{\boldsymbol{K}}\boldsymbol{S} - \boldsymbol{S}^\mathsf{T}\boldsymbol{K}\boldsymbol{S})) - \operatorname{tr}((\boldsymbol{S}^\mathsf{T}\hat{\boldsymbol{K}}\boldsymbol{S})^{-1}\boldsymbol{S}^\mathsf{T}\boldsymbol{K}\boldsymbol{S})\right)$$

$$= \frac{1}{2}\left(\tilde{\boldsymbol{v}}_i^\mathsf{T} \boldsymbol{S}^\mathsf{T} \boldsymbol{K}\boldsymbol{S}\tilde{\boldsymbol{v}}_i - \log\det((\boldsymbol{S}^\mathsf{T}\hat{\boldsymbol{K}}\boldsymbol{S})^{-1}(\sigma^2\boldsymbol{S}^\mathsf{T}\boldsymbol{S})) - \operatorname{tr}((\boldsymbol{S}^\mathsf{T}\hat{\boldsymbol{K}}\boldsymbol{S})^{-1}\boldsymbol{S}^\mathsf{T}\boldsymbol{K}\boldsymbol{S})\right)$$

$$= \frac{1}{2}\left(\tilde{\boldsymbol{v}}_i^\mathsf{T} \boldsymbol{S}^\mathsf{T} \boldsymbol{K}\boldsymbol{S}\tilde{\boldsymbol{v}}_i + \log\det(\boldsymbol{S}^\mathsf{T}\hat{\boldsymbol{K}}\boldsymbol{S}) - \log\det(\sigma^2\boldsymbol{S}^\mathsf{T}\boldsymbol{S}) - \operatorname{tr}((\boldsymbol{S}^\mathsf{T}\hat{\boldsymbol{K}}\boldsymbol{S})^{-1}\boldsymbol{S}^\mathsf{T}\boldsymbol{K}\boldsymbol{S})\right)$$

$$= \frac{1}{2}\left(\tilde{\boldsymbol{v}}_i^\mathsf{T} \boldsymbol{S}^\mathsf{T} \boldsymbol{K}\boldsymbol{S}\tilde{\boldsymbol{v}}_i + \log\det(\boldsymbol{S}^\mathsf{T}\hat{\boldsymbol{K}}\boldsymbol{S}) - i\log(\sigma^2) - \log\det(\boldsymbol{S}^\mathsf{T}\boldsymbol{S}) - \operatorname{tr}((\boldsymbol{S}^\mathsf{T}\hat{\boldsymbol{K}}\boldsymbol{S})^{-1}\boldsymbol{S}^\mathsf{T}\boldsymbol{K}\boldsymbol{S})\right)$$

$$\square$$

## S2.3    Comparison of Training Losses

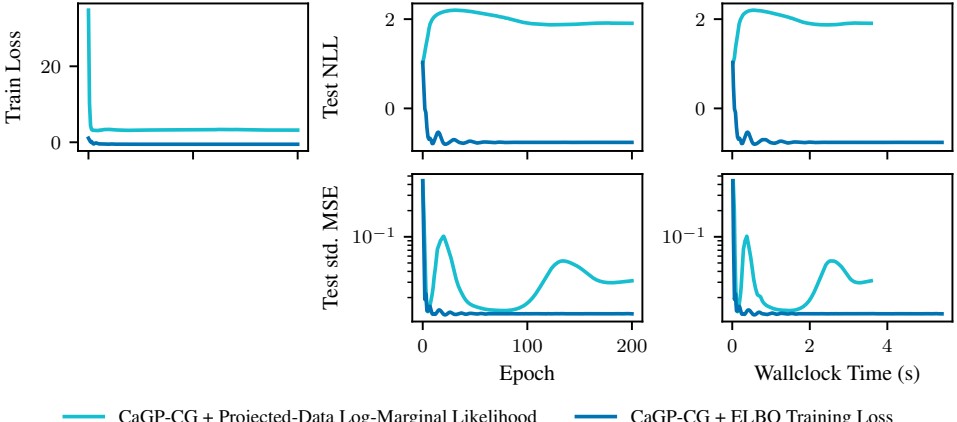

**Figure S1:** *Comparison of two different training losses for CaGP.* The naive choice of the projected-data log-marginal likelihood leads to increasingly worse generalization performance as measured by NLL. In comparison, the ELBO training loss leads to much better performance.

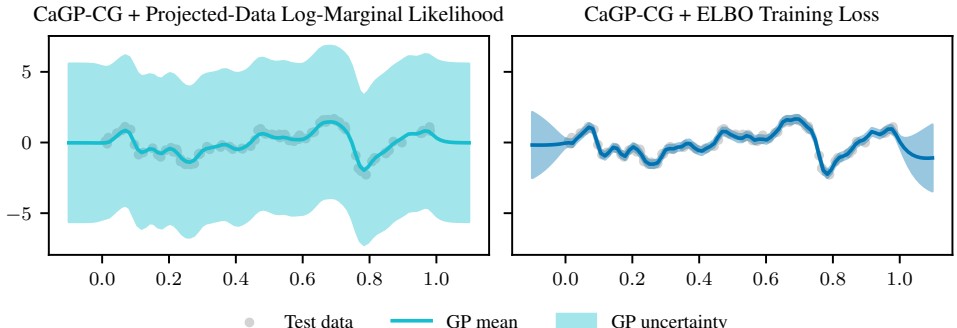

**Figure S2:** *CaGP predictive distributions with hyperparameters optimized using different losses.* When optimizing hyperparameters with respect to the projected-data log-marginal likelihood, CaGP-CG completely overestimates the noise scale, which leads to increasingly worse generalization performance. In comparison, the ELBO training loss leads to a much better overall fit.

## S3    Choice of Actions

We begin by proving that the *CaGP posterior in Equation* (5) *is uniquely defined by the space spanned by the columns of the actions* $\mathrm{colsp}(\boldsymbol{S})$, rather than the specific choice of the matrix $\boldsymbol{S}$.

**Lemma S4** (The CaGP Posterior Is Uniquely Defined by the Column Space of the Actions)
*Let $\boldsymbol{S}, \boldsymbol{S}' \in \mathbb{R}^{n \times i}$ be two action matrices, each of which consists of non-zero and linearly independent action vectors, such that their column spaces are identical, i.e.*

$$\mathrm{colsp}(\boldsymbol{S}) = \mathrm{colsp}(\boldsymbol{S}'), \tag{S20}$$

*then the corresponding CaGP posteriors $\mathcal{GP}(\mu_i, K_i)$ and $\mathcal{GP}(\mu_i', K_i')$ are equivalent.*

*Proof.* By assumption (S20) there exists $\boldsymbol{W} \in \mathbb{R}^{i \times i}$ such that $\boldsymbol{S}' = \boldsymbol{SW}$. Since action vectors are assumed to be linearly independent and non-zero, it holds that $i = \mathrm{rank}(\boldsymbol{S}') = \mathrm{rank}(\boldsymbol{SW}) = \mathrm{rank}(\boldsymbol{W})$, where the last equality follows from standard properties of the matrix rank. Therefore $\boldsymbol{W}$ is invertible, and we have that

$$\boldsymbol{C}' = \boldsymbol{S}'(\boldsymbol{S}'^{\mathsf{T}}\hat{\boldsymbol{K}}\boldsymbol{S}')^{-1}\boldsymbol{S}'^{\mathsf{T}} = \boldsymbol{SW}(\boldsymbol{W}^{\mathsf{T}}\boldsymbol{S}^{\mathsf{T}}\hat{\boldsymbol{K}}\boldsymbol{SW})^{-1}\boldsymbol{W}^{\mathsf{T}}\boldsymbol{S}^{\mathsf{T}} = \boldsymbol{S}(\boldsymbol{S}^{\mathsf{T}}\hat{\boldsymbol{K}}\boldsymbol{S})^{-1}\boldsymbol{S}^{\mathsf{T}} = \boldsymbol{C}.$$

Since the CaGP posterior in Equation (5) is fully defined via the approximate precision matrix $\boldsymbol{C}$, the desired result follows. $\qquad\square$

**Corollary S1** (Action Order and Magnitude Does Not Change CaGP Posterior)
*The CaGP posterior in Equation* (5) *is invariant under permutation and rescaling of the actions.*

*Proof.* This follows immediately by choosing a permutation or a diagonal matrix $\boldsymbol{W}$, respectively, such that $\boldsymbol{S}' = \boldsymbol{S}\boldsymbol{W}$ in Lemma S4. $\qquad\square$

### S3.1 (Conjugate) Gradient / Residual Policy

Consider the following linear system

$$\hat{\boldsymbol{K}}\boldsymbol{v}_\star = \boldsymbol{y} - \boldsymbol{\mu} \tag{S21}$$

with symmetric positive definite kernel matrix $\hat{\boldsymbol{K}} = \boldsymbol{K} + \sigma^2\boldsymbol{I}$, observations $\boldsymbol{y}$, prior mean evaluated at the data $\boldsymbol{\mu} = \mu(\boldsymbol{X})$ and representer weights $\boldsymbol{v}_\star$.

**Lanczos process [23]**   The Lanczos process is an iterative method, which computes approximate eigenvalues $\hat{\boldsymbol{\Lambda}} = \mathrm{diag}(\hat{\lambda}_1, \dots, \hat{\lambda}_i) \in \mathbb{R}^{i\times i}$ and approximate eigenvectors $\hat{\boldsymbol{U}} = (\hat{\boldsymbol{u}}_1 \cdots \hat{\boldsymbol{u}}_i) \in \mathbb{R}^{n\times i}$ for a symmetric positive definite matrix $\hat{\boldsymbol{K}}$ by repeated matrix-vector multiplication. Given an arbitrary starting vector $\boldsymbol{q}_1 \in \mathbb{R}^n$, s.t. $\|\boldsymbol{q}_1\|_2 = 1$, it returns $i$ orthonormal vectors $\boldsymbol{Q} = (\boldsymbol{q}_1 \cdots \boldsymbol{q}_i) \in \mathbb{R}^{n\times i}$ and a tridiagonal matrix $\boldsymbol{T} = \boldsymbol{Q}^\mathsf{T}\hat{\boldsymbol{K}}\boldsymbol{Q} \in \mathbb{R}^{i\times i}$. The eigenvalue approximations are given by an eigendecomposition of $\boldsymbol{T} = \boldsymbol{W}\hat{\boldsymbol{\Lambda}}\boldsymbol{W}^\mathsf{T}$, where $\boldsymbol{W} \in \mathbb{R}^{i\times i}$ orthonormal, and the eigenvector approximations are then given by $\hat{\boldsymbol{U}} = \boldsymbol{Q}\boldsymbol{W} \in \mathbb{R}^{n\times i}$ [e.g. Sec. 10.1.4 of 62].

**Conjugate Gradient Method [24]**   The conjugate gradient method is an iterative method to solve linear systems with symmetric positive definite system matrix by repeated matrix-vector multiplication. When applied to Equation (S21), it produces a sequence of representer weights approximations $\boldsymbol{v}_i \approx \boldsymbol{v}_\star = \hat{\boldsymbol{K}}^{-1}(\boldsymbol{y} - \boldsymbol{\mu})$. Its residuals $\boldsymbol{r}_i = \boldsymbol{y} - \boldsymbol{\mu} - \hat{\boldsymbol{K}}\boldsymbol{v}_i$ are proportional to the Lanczos vectors for a Lanczos process initialized at $\boldsymbol{q}_1 = \frac{\boldsymbol{r}_0}{\|\boldsymbol{r}_0\|_2}$, i.e. $\boldsymbol{Q} = \boldsymbol{R}\boldsymbol{D}$ where $\boldsymbol{R} \in \mathbb{R}^{n\times i}$ is the matrix of residuals and $\boldsymbol{D} \in \mathbb{R}^{i\times i}$ a diagonal matrix (e.g. [Alg. 11.3.2 in 62] or [Sec. 3 & Eqn. (3.4) of 63]).

Therefore choosing actions defined by the residuals of CG in CaGP-CG, i.e. $\boldsymbol{S} = \boldsymbol{R}$, is equivalent to choosing actions $\boldsymbol{S}' = \hat{\boldsymbol{U}}$ given by the eigenvector approximations computed by the Lanczos process initialized as above, since

$$\mathrm{colsp}(\boldsymbol{S}) = \mathrm{colsp}(\boldsymbol{R}) = \mathrm{colsp}(\boldsymbol{R}\boldsymbol{D}) = \mathrm{colsp}(\boldsymbol{Q}) = \mathrm{colsp}(\boldsymbol{Q}\boldsymbol{W}) = \mathrm{colsp}(\hat{\boldsymbol{U}}) = \mathrm{colsp}(\boldsymbol{S}')$$

and by Lemma S4 it holds that the corresponding CaGP posteriors with actions $\boldsymbol{S}$ and $\boldsymbol{S}'$ are equivalent.

### S3.2 Information-theoretic Policy

In information-theoretic formulations of active learning, new data is selected to minimize uncertainty about a set of latent variables $\boldsymbol{z}$. In other words, we would aim to minimize the entropy of the posterior $\mathrm{H}_{p(\boldsymbol{z}|\boldsymbol{X})}(\boldsymbol{z}) = -\int \log p(\boldsymbol{z} \mid \boldsymbol{X})p(\boldsymbol{z} \mid \boldsymbol{X})\, d\boldsymbol{z}$ as a function of the data $\boldsymbol{X}$ [21]. In analogy to active learning, in our setting we propose to perform computations $\boldsymbol{y} \mapsto \boldsymbol{S}_i^\mathsf{T}\boldsymbol{y}$ to maximally reduce uncertainty about the latent function $f(\boldsymbol{X})$ evaluated at the training data.

**Lemma S5** (Information-theoretic Policy)
*The actions $\boldsymbol{S}$ minimizing the entropy of the computation-aware posterior $p(f(\boldsymbol{X}) \mid \boldsymbol{S}^\mathsf{T}\boldsymbol{y})$ at the training data, or equivalently the actions maximizing the mutual information between $f(\boldsymbol{X})$ and the projected data $\boldsymbol{S}^\mathsf{T}\boldsymbol{y}$, are given by*

$$(\boldsymbol{s}_1, \dots, \boldsymbol{s}_i) = \operatorname*{arg\,min}_{\boldsymbol{S}\in\mathbb{R}^{n\times i}} \mathrm{H}_{p(f(\boldsymbol{X})|\boldsymbol{S}^\mathsf{T}\boldsymbol{y})}(f(\boldsymbol{X})) \tag{S22}$$

$$= \operatorname*{arg\,max}_{\boldsymbol{S}\in\mathbb{R}^{n\times i}} \underbrace{\mathrm{H}(f(\boldsymbol{X})) - \mathrm{H}\big(f(\boldsymbol{X}) \mid \boldsymbol{S}^\mathsf{T}\boldsymbol{y}\big)}_{=:\mathrm{MI}(f(\boldsymbol{X});\boldsymbol{S}^\mathsf{T}\boldsymbol{y})} \tag{S23}$$

*where $\boldsymbol{s}_1, \dots, \boldsymbol{s}_i$ are the top-$i$ eigenvectors of $\hat{\boldsymbol{K}}$ in descending order of the eigenvalue magnitude.*

*Proof.* Let $\tilde{\boldsymbol{y}} := \boldsymbol{S}^\mathsf{T}\boldsymbol{y}$ and $\boldsymbol{f} := f(\boldsymbol{X})$. By assumption, we have that $\boldsymbol{f} \sim \mathcal{N}(\boldsymbol{\mu}, \boldsymbol{K})$. Recall that the entropy of a Gaussian random vector $\boldsymbol{f} \sim \mathcal{N}(\boldsymbol{m}, \boldsymbol{S})$ is given by $\mathrm{H}(\boldsymbol{f}) = \frac{1}{2}\big(\log\det(\boldsymbol{S}) + n\log(2\pi e)\big)$. Now since the covariance function of the computation-aware posterior in Equation (5) does not depend on the targets $\boldsymbol{y}$, neither does its entropy $\mathrm{H}_{p(\boldsymbol{f}|\boldsymbol{S}^\mathsf{T}\boldsymbol{y})}(\boldsymbol{f})$.

Therefore, by definition of the *conditional* entropy and using the law of the unconscious statistician, it holds that

$$
\mathrm{H}(\boldsymbol{f} \mid \tilde{\boldsymbol{y}}) = -\int\int \log p(\boldsymbol{f}\mid\tilde{\boldsymbol{y}})p(\boldsymbol{f}\mid\tilde{\boldsymbol{y}})p(\tilde{\boldsymbol{y}})\,d\boldsymbol{f}\,d\tilde{\boldsymbol{y}}
$$
$$
= \mathbb{E}_{p(\tilde{\boldsymbol{y}})}\big(\mathrm{H}_{p(\boldsymbol{f}|\tilde{\boldsymbol{y}})}(\boldsymbol{f})\big)
$$
$$
= \mathbb{E}_{p(\boldsymbol{y})}\big(\mathrm{H}_{p(\boldsymbol{f}|\boldsymbol{S}^\mathsf{T}\boldsymbol{y})}(\boldsymbol{f})\big)
$$

and since the covariance of a Gaussian conditioned on data doesn't depend on the data, we have

$$
= \mathrm{H}_{p(\boldsymbol{f}|\boldsymbol{S}^\mathsf{T}\boldsymbol{y})}(\boldsymbol{f})
$$

Therefore we can rewrite the mutual information in terms of prior and posterior entropy, such that

$$
\mathrm{H}(\boldsymbol{f}) - \mathrm{H}\big(\boldsymbol{f}\mid\boldsymbol{S}^\mathsf{T}\boldsymbol{y}\big) = \mathrm{H}(\boldsymbol{f}) - \mathrm{H}_{p(\boldsymbol{f}|\boldsymbol{S}^\mathsf{T}\boldsymbol{y})}(\boldsymbol{f})
$$
$$
= \frac{1}{2}\big(\log\det(\boldsymbol{K}) + n\log(2\pi e) - \log\det(\boldsymbol{K} - \boldsymbol{K}\boldsymbol{C}_i\boldsymbol{K}) - n\log(2\pi e)\big)
$$
$$
= -\frac{1}{2}\log\Big(\det(\boldsymbol{K} - \boldsymbol{K}\boldsymbol{S}(\boldsymbol{S}^\mathsf{T}\hat{\boldsymbol{K}}\boldsymbol{S})^{-1}\boldsymbol{S}^\mathsf{T}\boldsymbol{K})\det(\boldsymbol{K}^{-1})\Big)
$$

Via the matrix determinant lemma $\det(\boldsymbol{A}+\boldsymbol{U}\boldsymbol{W}\boldsymbol{V}^\mathsf{T}) = \det(\boldsymbol{W}^{-1}+\boldsymbol{V}^\mathsf{T}\boldsymbol{A}^{-1}\boldsymbol{U})\det(\boldsymbol{W})\det(\boldsymbol{A})$, we obtain

$$
= -\frac{1}{2}\log\Big(\det(-\boldsymbol{S}^\mathsf{T}\hat{\boldsymbol{K}}\boldsymbol{S} + \boldsymbol{S}\boldsymbol{K}\boldsymbol{K}^{-1}\boldsymbol{K})\det(-(\boldsymbol{S}^\mathsf{T}\hat{\boldsymbol{K}}\boldsymbol{S})^{-1})\det(\boldsymbol{K})\det(\boldsymbol{K}^{-1})\Big)
$$
$$
= -\frac{1}{2}\log\Big(\det(\boldsymbol{S}^\mathsf{T}\boldsymbol{K}\boldsymbol{S} - \boldsymbol{S}^\mathsf{T}\hat{\boldsymbol{K}}\boldsymbol{S})\det(-(\boldsymbol{S}^\mathsf{T}\hat{\boldsymbol{K}}\boldsymbol{S})^{-1})\Big)
$$
$$
= -\frac{1}{2}\log\det(\sigma^2\boldsymbol{S}^\mathsf{T}\boldsymbol{S}(\boldsymbol{S}^\mathsf{T}\hat{\boldsymbol{K}}\boldsymbol{S})^{-1})
$$
$$
= \frac{1}{2}\log\det(\sigma^{-2}(\boldsymbol{S}^\mathsf{T}\boldsymbol{S})^{-1}\boldsymbol{S}^\mathsf{T}\hat{\boldsymbol{K}}\boldsymbol{S})
$$
$$
= \frac{1}{2}\big(\log\det((\boldsymbol{S}^\mathsf{T}\boldsymbol{S})^{-1}\boldsymbol{S}^\mathsf{T}\hat{\boldsymbol{K}}\boldsymbol{S}) - i\log(\sigma^2)\big)
$$
$$
= \frac{1}{2}\big(\log\det(\boldsymbol{L}^{-\mathsf{T}}\boldsymbol{S}^\mathsf{T}\hat{\boldsymbol{K}}\boldsymbol{S}\boldsymbol{L}^{-1}) - i\log(\sigma^2)\big)
$$

for $\boldsymbol{L}$ a square root of $\boldsymbol{S}^\mathsf{T}\boldsymbol{S}$. Now we can upper bound the above as follows

$$
\max_{\boldsymbol{S}\in\mathbb{R}^{n\times i}} \mathrm{H}(\boldsymbol{f}) - \mathrm{H}\big(\boldsymbol{f}\mid\boldsymbol{S}^\mathsf{T}\boldsymbol{y}\big) \leq \max_{\tilde{\boldsymbol{S}}\in\mathbb{R}^{n\times i}} \frac{1}{2}\big(\log\det(\tilde{\boldsymbol{S}}^\mathsf{T}\hat{\boldsymbol{K}}\tilde{\boldsymbol{S}}) - i\log(\sigma^2)\big)
$$
$$
= \frac{1}{2}\big(\log\det(\boldsymbol{U}\hat{\boldsymbol{K}}\boldsymbol{U}^\mathsf{T}) - i\log(\sigma^2)\big)
$$
$$
= \frac{1}{2}\Big(\sum_{j=1}^{i}\log(\lambda_j(\hat{\boldsymbol{K}})) - i\log(\sigma^2)\Big)
$$

where $\boldsymbol{U}$ are the orthonormal eigenvectors of $\hat{\boldsymbol{K}}$ for the largest $i$ eigenvalues. Now choosing $\boldsymbol{S} = \boldsymbol{U}$ achieves the upper bound since $\boldsymbol{U}^\mathsf{T}\boldsymbol{U} = \boldsymbol{I}$ and therefore $\boldsymbol{S}_i = \boldsymbol{U}$ is a solution to the optimization problem.

Finally using the argument above and since $\mathrm{H}(\boldsymbol{f})$ does not depend on $\boldsymbol{S}$, we have that

$$
\arg\max_{\boldsymbol{S}\in\mathbb{R}^{n\times i}} \mathrm{H}(\boldsymbol{f}) - \mathrm{H}\big(\boldsymbol{f}\mid\boldsymbol{S}^\mathsf{T}\boldsymbol{y}\big) = \arg\max_{\boldsymbol{S}\in\mathbb{R}^{n\times i}} \mathrm{H}(\boldsymbol{f}) - \mathrm{H}_{p(\boldsymbol{f}|\boldsymbol{S}^\mathsf{T}\boldsymbol{y})}(\boldsymbol{f}) = \arg\min_{\boldsymbol{S}\in\mathbb{R}^{n\times i}} \mathrm{H}_{p(\boldsymbol{f}|\boldsymbol{S}^\mathsf{T}\boldsymbol{y})}(\boldsymbol{f}).
$$

This proves the claim. $\qquad\square$

# S4  Algorithms

## S4.1  Iterative and Batch Versions of CaGP

---

Algorithm S1: CaGP = IterGP: Iterative formulation as in Wenger et al. [19]

---

**Input:** GP prior $\mathcal{GP}(\mu, K)$, training data $(\boldsymbol{X}, \boldsymbol{y})$
**Output:** (combined) GP posterior $\mathcal{GP}(\mu_i, K_i)$

| | | | Time | Space |
|---|---|---|---|---|
| 1 | **procedure** CAGP$(\mu, K, \boldsymbol{X}, \boldsymbol{y}, \boldsymbol{C}_0 = \boldsymbol{0})$ | | Time | Space |
| 2 | **while not** STOPPINGCRITERION() **do** | | | |
| 3 | $\boldsymbol{s}_i \leftarrow$ POLICY() | Select action via policy. | | |
| 4 | $\boldsymbol{r}_{i-1} \leftarrow (\boldsymbol{y} - \boldsymbol{\mu}) - \hat{\boldsymbol{K}}\boldsymbol{v}_{i-1}$ | Residual. | $\mathcal{O}(n^2)$ | $\mathcal{O}(n)$ |
| 5 | $\alpha_i \leftarrow \boldsymbol{s}_i^\mathsf{T}\boldsymbol{r}_{i-1}$ | Observation. | $\mathcal{O}(k)$ | $\mathcal{O}(1)$ |
| 6 | $\boldsymbol{z}_i \leftarrow \hat{\boldsymbol{K}}\boldsymbol{s}_i$ | | $\mathcal{O}(nk)$ | $\mathcal{O}(n)$ |
| 7 | $\boldsymbol{d}_i \leftarrow \Sigma_{i-1}\hat{\boldsymbol{K}}\boldsymbol{s}_i = \boldsymbol{s}_i - \boldsymbol{C}_{i-1}\boldsymbol{z}_i$ | Search direction. | $\mathcal{O}(ni)$ | $\mathcal{O}(n)$ |
| 8 | $\eta_i \leftarrow \boldsymbol{s}_i^\mathsf{T}\hat{\boldsymbol{K}}\Sigma_{i-1}\hat{\boldsymbol{K}}\boldsymbol{s}_i = \boldsymbol{z}_i^\mathsf{T}\boldsymbol{d}_i$ | | $\mathcal{O}(n)$ | $\mathcal{O}(1)$ |
| 9 | $\boldsymbol{C}_i \leftarrow \boldsymbol{C}_{i-1} + \frac{1}{\eta_i}\boldsymbol{d}_i\boldsymbol{d}_i^\mathsf{T}$ | Precision matrix approx. $\boldsymbol{C}_i \approx \hat{\boldsymbol{K}}^{-1}$. | $\mathcal{O}(n)$ | $\mathcal{O}(ni)$ |
| 10 | $\boldsymbol{v}_i \leftarrow \boldsymbol{v}_{i-1} + \frac{\alpha_i}{\eta_i}\boldsymbol{d}_i$ | Representer weights estimate. | $\mathcal{O}(n)$ | $\mathcal{O}(n)$ |
| 11 | $\Sigma_i \leftarrow \Sigma_0 - \boldsymbol{C}_i$ | Representer weights uncertainty. | | |
| 12 | $\mu_i(\cdot) \leftarrow \mu(\cdot) + K(\cdot, \boldsymbol{X})\boldsymbol{v}_i$ | Approximate posterior mean. | $\mathcal{O}(n_\diamond n)$ | $\mathcal{O}(n_\diamond)$ |
| 13 | $K_i(\cdot, \cdot) \leftarrow K(\cdot, \cdot) - K(\cdot, \boldsymbol{X})\boldsymbol{C}_i K(\boldsymbol{X}, \cdot)$ | Combined covariance function. | $\mathcal{O}(n_\diamond ni)$ | $\mathcal{O}(n_\diamond^2)$ |
| 14 | **return** $\mathcal{GP}(\mu_i, K_i)$ | | | |

---

---

Algorithm S2: CaGP: Batch Version

---

**Input:** GP prior $\mathcal{GP}(\mu, K)$, training data $(\boldsymbol{X}, \boldsymbol{y})$
**Output:** (combined) GP posterior $\mathcal{GP}(\mu_i, K_i)$

| | | | Time | Space |
|---|---|---|---|---|
| 1 | **procedure** CAGP$(\mu, K, \boldsymbol{X}, \boldsymbol{y})$ | | Time | Space |
| 2 | $\boldsymbol{S}_i \leftarrow$ POLICY() | Select batch of actions via policy. | | |
| 3 | $\tilde{\boldsymbol{y}} \leftarrow \boldsymbol{S}_i^\mathsf{T}(\boldsymbol{y} - \boldsymbol{\mu})$ | "Projected" data. | $\mathcal{O}(ki)$ | $\mathcal{O}(i)$ |
| 4 | $\boldsymbol{Z}_i \leftarrow \hat{\boldsymbol{K}}\boldsymbol{S}_i$ | | $\mathcal{O}(nki)$ | $\mathcal{O}(ni)$ |
| 5 | $\boldsymbol{L}_i \leftarrow$ CHOLESKY$(\boldsymbol{S}_i^\mathsf{T}\boldsymbol{Z}_i)$ | | $\mathcal{O}(i^2(i + k))$ | $\mathcal{O}(i^2)$ |
| 6 | $\tilde{\boldsymbol{v}}_i \leftarrow \boldsymbol{L}_i^{-\mathsf{T}}\boldsymbol{L}_i^{-1}\tilde{\boldsymbol{y}}$ | "Projected" representer weights. | $\mathcal{O}(i^2)$ | $\mathcal{O}(i)$ |
| 7 | $K_{\boldsymbol{S}}(\cdot, \boldsymbol{X}) \leftarrow K(\cdot, \boldsymbol{X})\boldsymbol{S}_i$ | | $\mathcal{O}(n_\diamond ki)$ | $\mathcal{O}(n_\diamond i)$ |
| 8 | $\mu_i(\cdot) \leftarrow \mu(\cdot) + K_{\boldsymbol{S}}(\cdot, \boldsymbol{X})\tilde{\boldsymbol{v}}_i$ | | $\mathcal{O}(n_\diamond i)$ | $\mathcal{O}(n_\diamond)$ |
| 9 | $K_i(\cdot, \cdot) \leftarrow K(\cdot, \cdot) - K_{\boldsymbol{S}}(\cdot, \boldsymbol{X})\boldsymbol{L}_i^{-\mathsf{T}}\boldsymbol{L}_i^{-1}K_{\boldsymbol{S}}(\boldsymbol{X}, \cdot)$ | | $\mathcal{O}(n_\diamond i^2)$ | $\mathcal{O}(n_\diamond^2)$ |
| 10 | **return** $\mathcal{GP}(\mu_i, K_i)$ | | | |

---

## S4.2  Implementation

We provide an open-source implementation of CaGP-Opt as part of GPyTorch. To install the package via `pip`, execute the following in the command line:

```
pip install git+https://github.com/cornellius-gp/linear_operator.git@sparsity
pip install git+https://github.com/cornellius-gp/gpytorch.git@computation-aware-gps-v2
pip install pykeops
```

## S5 Additional Experimental Results and Details

### S5.1 Inducing Points Placement and Uncertainty Quantification of SVGP

To better understand whether the overconfidence of SVGP at inducing points observed in the visualization in Figure 1 holds also in higher dimensions, we do the following experiment. For varying input dimension $d \in \{1, 2, \ldots, 25\}$, we generate synthetic training data by sampling $n = 500$ inputs $\boldsymbol{X}$ uniformly at random with corresponding targets sampled from a zero-mean Gaussian process $y \sim \mathcal{GP}(0, K^\sigma)$, where $K^\sigma(\cdot, \cdot) = K(\cdot, \cdot) + \sigma^2 \delta(\cdot, \cdot)$ is given by the sum of a Matérn($3/2$) and a white noise kernel with noise scale $\sigma$. We optimize the kernel hyperparameters, variational parameters and inducing points ($m = 64$) jointly for 300 epochs using Adam with a linear learning rate scheduler. At convergence we measure the *average distance between inducing points and the nearest datapoint measured in lengthscale units*, i.e.

$$\bar{d}_{\boldsymbol{l}}(\boldsymbol{Z}, \boldsymbol{X}) = \frac{1}{m} \sum_{i=1}^{m} \left( \min_j \|\boldsymbol{z}_i - \boldsymbol{x}_j\|_{\mathrm{diag}(\boldsymbol{l}^{-2})} \right) \tag{S24}$$

where $\boldsymbol{l} \in \mathbb{R}^d$ is the vector of lengthscales (one per input dimension). We also compute the *average ratio of the posterior variance to the predictive variance at the inducing points*, i.e.

$$\bar{\rho}(\boldsymbol{Z}) = \frac{1}{m} \sum_{i=1}^{m} \frac{K_{\mathrm{posterior}}(\boldsymbol{z}_i, \boldsymbol{z}_i)}{K_{\mathrm{posterior}}(\boldsymbol{z}_i, \boldsymbol{z}_i) + \sigma^2}. \tag{S25}$$

The results of our experiments are shown in Figure S3. We find that as expected the inducing points are optimized to lie closer to datapoints than points sampled uniformly at random. However, the inducing points lie increasingly far away from the training data as the dimension increases relative to the lengthscale that SVGP learns. Therefore this experiment suggests that the phenomenon observed in Figure 1, that SVGP can be overconfident at inducing points if they are far away from training datapoints, to be increasingly present as the input dimension increases. This is further substantiated by Figure S3(b) since the proportion of posterior variance to predictive variance at the inducing points is very small already in $d = 4$ dimensions. This illustrates both SVGP's overconfidence at the inducing points (in particular in higher dimensions) and that its predictive variance is dominated by the learned observation noise, as we also saw in the illustrative Figure 1.

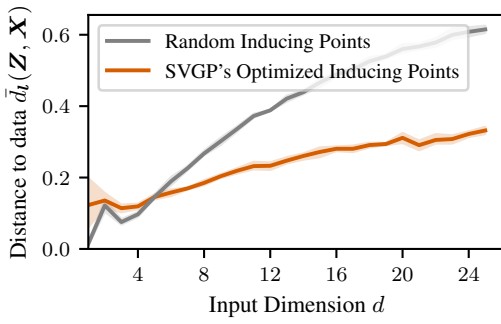 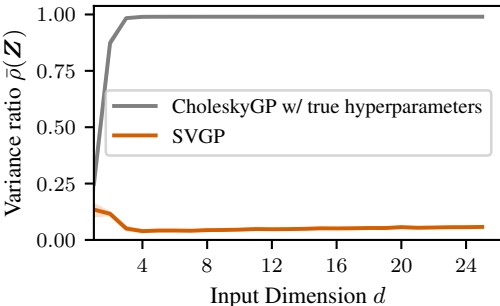

(a) Average distance of inducing points to the nearest datapoint measured in lengthscale units.

(b) Average ratio of posterior to predictive variance at SVGP's inducing point locations.

Figure S3: *SVGP's inducing point placement and uncertainty in higher dimensions.* (a) As the dimension increases, the inducing points SVGP learns lie increasingly far away from the data measured in lengthscale units given a fixed training data set size and number of inducing points. (b) SVGP's variance at the inducing points is dominated by the learned observational noise in higher dimensions, rather than by the posterior variance. The comparison to a CholeskyGP with the data-generating hyperparameters shows that SVGP compensates for a lack of posterior variance at the inducing points by artificially inflating the observation noise. This illustrates both the overconfidence (in terms of posterior variance) of SVGP at the inducing points and its tendency to oversmooth.

## S5.2 Grassman Distance Between Subspaces

In Figure 2 we compute the distance between the subspaces spanned by random vectors, the actions $\boldsymbol{S}$ of CaGP, and the space spanned by the top-$i$ eigenvectors. The notion of subspace distance we use is the Grassman distance, i.e. for two subspaces spanned by the columns of matrices $\boldsymbol{A} \in \mathbb{R}^{n \times p}$ and $\boldsymbol{B} \in \mathbb{R}^{n \times p}$ s.t. $p \geq q$ the Grassman subspace distance is defined by

$$d(\boldsymbol{A}, \boldsymbol{B}) = \|\boldsymbol{\theta}\|_2 \tag{S26}$$

where $\boldsymbol{\theta} \in \mathbb{R}^q$ is the vector of principal angles between the two spaces, which can be computed via an SVD [e.g. Alg. 6.4.3 in 62].

## S5.3 Generalization Experiment

Table S1: Detailed configuration of the generalization experiment in Section 5.

| Method | Posterior Approximation | | Model Selection / Training | | | | |
| | Iters. $i$ / Ind. Points $m$ | Solver Tol. | Optimizer | Epochs | (Initial) Learning Rate | Batch Size | Precision |
|---|---|---|---|---|---|---|---|
| CholeskyGP | - | - | LBFGS | 100 | $\{1, 10^{-1}, 10^{-2}, 10^{-3}, 10^{-4}\}$ | $n$ | float64 |
| SGPR | 1024 | - | Adam | 1000 | $\{1, 10^{-1}, 10^{-2}, 10^{-3}, 10^{-4}\}$ | $n$ | float32 |
| | 1024 | - | LBFGS | 100 | $\{1, 10^{-1}, 10^{-2}, 10^{-3}, 10^{-4}\}$ | $n$ | float64 |
| SVGP | 1024 | - | Adam | 1000 | $\{1, 10^{-1}, 10^{-2}, 10^{-3}, 10^{-4}\}$ | 1024 | float32 |
| CGGP | 512 | $10^{-4}$ | LBFGS | 100 | $\{1, 10^{-1}, 10^{-2}, 10^{-3}, 10^{-4}\}$ | $n$ | float64 |
| CaGP-CG | 512 | $10^{-4}$ | Adam | 250 | $\{1, 10^{-1}\}$ | $n$ | float32 |
| CaGP-Opt | 512 | - | Adam | 1000 | $\{1, 10^{-1}, 10^{-2}\}$ | $n$ | float32 |
| | 512 | - | LBFGS | 100 | $\{1, 10^{-1}, 10^{-2}\}$ | $n$ | float64 |

### S5.3.1 Impact of Learning Rate on Generalization

To show the impact of different choices of learning rate on the GP approximations we consider, we show the test metrics for the learning rate sweeps in our main experiment in Figure S4. Note that not all choices of learning rate appear since a small minority of runs fail outright, for example if the learning rate is too large.

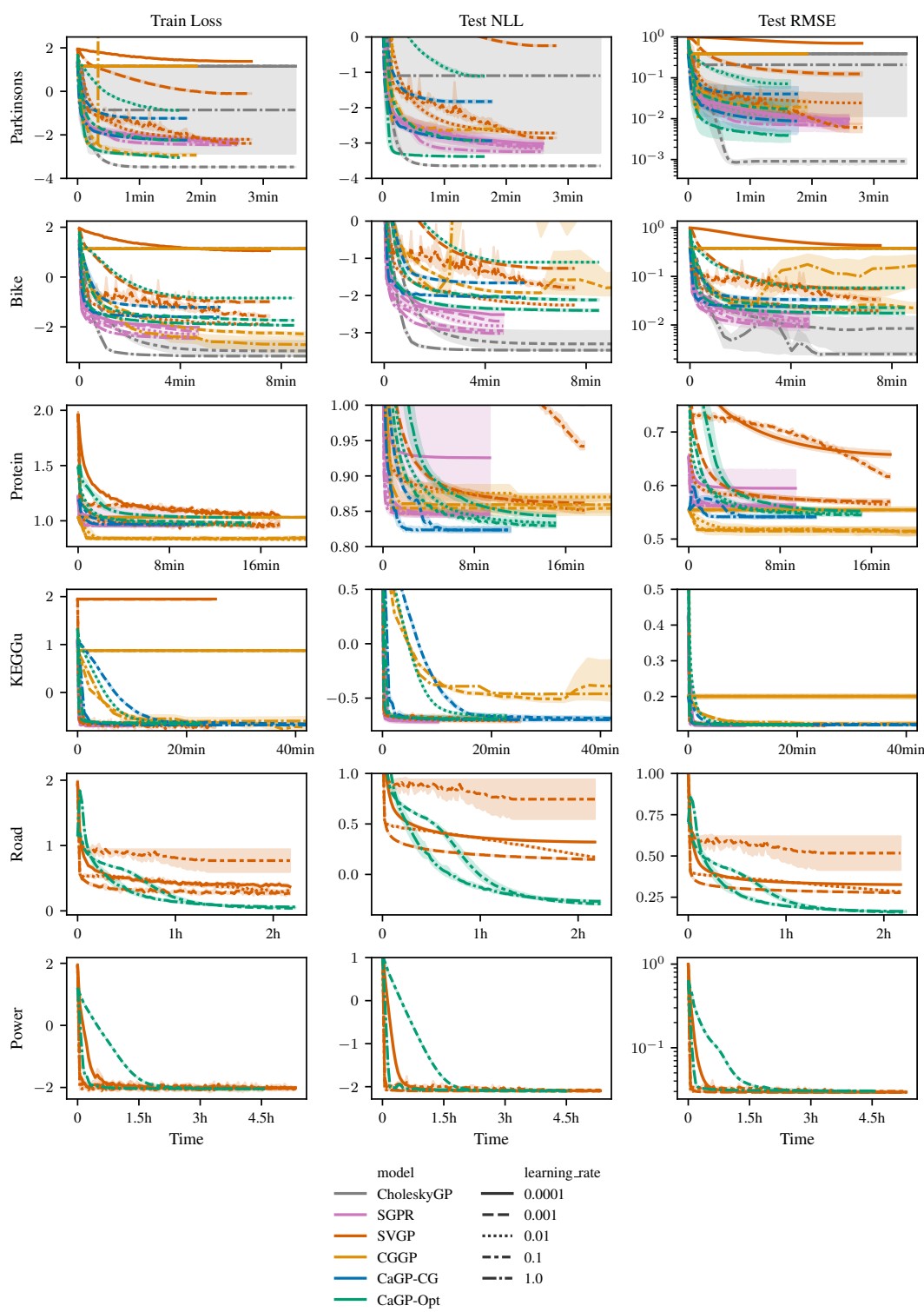

Figure S4: *Effects of (initial) learning rate when using either LBFGS with Wolfe line search (CholeskyGP, SGPR) or Adam (SVGP, CaGP-CG, CaGP-Opt) for hyperparameter optimization.*

## S5.3.2 Evolution Of Hyperparameters During Training

To better understand how the kernel hyperparameters of each method evolve during training, we show their trajectories in Figure S5 for each dataset. Note that we only show the first three lengthscales per dataset (rather than up to $d = 26$).

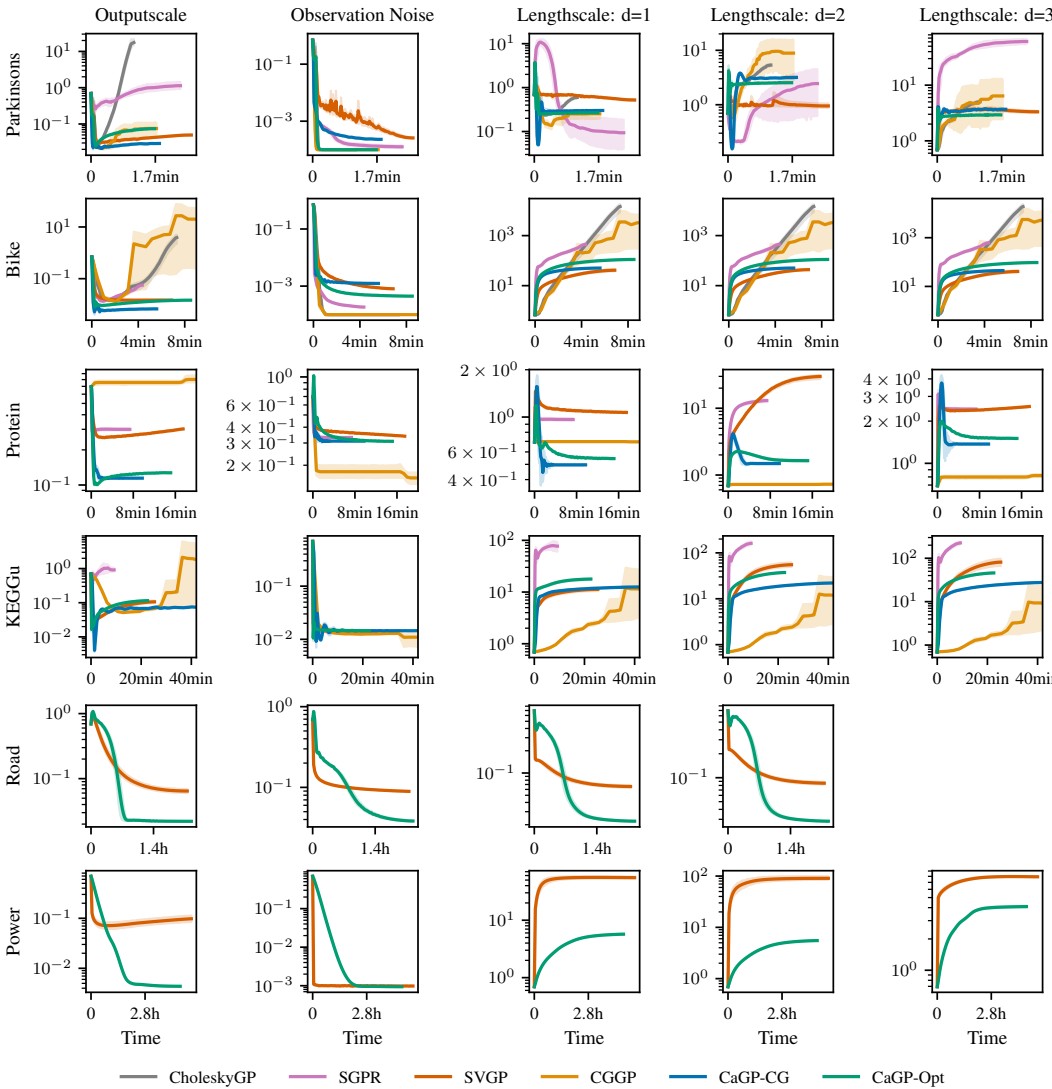

Figure S5: *Learned hyperparameters for different GP approximations on UCI datasets.* Showing only results for the best choice of learning rate per method.

