# OpenReview forum: "Computation-Aware Gaussian Processes: Model Selection And Linear-Time Inference"
_NeurIPS.cc/2024/Conference — NeurIPS 2024 poster_

### Official Review · Reviewer_wkY4 · 2024-06-22

**Soundness:** 2
**Presentation:** 3
**Contribution:** 3
**Rating:** 7
**Confidence:** 4

**Summary:**

The authors extend the work of Wenger et al. (2022) on "computational uncertainty" in Gaussian process regression models, which introduced the IterGP class of approximations. These approximations treat the limited computation in approximate GP methods as a source of uncertainty, resulting in uncertainty predictions that provably do not suffer from overconfidence. However, the original IterGP paper did not provide a straightforward method for model/hyperparameter selection: this paper addresses that by discussing two approaches for model selection, eventually settling on a new evidence lower bound using the variational family implied by the IterGP approximation, with different proposals for the "action matrix." They compare against SVGP and SGPR, two popular baselines for approximate Gaussian process regression.

**Strengths:**

The work presents a novel method for model selection and linear-time inference for the IterGP approximations originally proposed by Wenger et al. (2022). They discuss a number of viable options, and overall settle primarily on a newly derived ELBO with a sparse parameterization of an "action matrix," although they do also provide experiments for conjugate gradient-based "actions."

In my opinion, the quality of the work is largely high, is clearly written, and provides practical steps forward for the GP community. The paper is enjoyable to read and it was great to understand the various avenues that the authors explored to achieve practical model selection. Generally, the authors also seem relatively open about the limitations of the method and don't try to over-claim what they have achieved.

**Weaknesses:**

My main concerns with the paper concern its in its evaluation, particularly with regards to the sparse variational methods. I do not recall seeing these problems on this dataset, although I have admittedly found Parkinson's more problematic, but not in the way shown (I usually find that it just fails with a bad Cholesky decomposition instead of showing divergent behavior).

In my experience, this is because while GPyTorch is a state-of-the-art package for conjugate gradient-based methods, it is problematic when it comes to its implementations of sparse variational methods. I believe these instabilities are rather due to GPyTorch's use of Lanczos and conjugate gradient algorithms when computing covariance matrix decompositions and log probabilities, in addition to the use of float32. In my experience, stochastic variational methods are very sensitive to numerical accuracy and so should only be computed using the full Cholesky decomposition and float64. Therefore, since I did not see mention of this (although apologies if I have missed it!) I would recommend changing the GPyTorch defaults to use Cholesky and float64, or using a package that does not rely on approximate matrix decompositions such as GPflow or GPJax.

Moreover, one of the key attributes of SGPR is that because the loss is not stochastic, it can be optimized (for both hyperparameters and inducing locations) using second-order optimizers such as L-BFGS-B, as is commonly done. This is typically much faster than Adam, and essentially allows optimization to be parameter-free (as the usual defaults generally work well). It would have been good to compare to this, as is common in the literature, although I do commend the parameter search that the authors did for Adam optimization.

Finally, it would be interesting to see how the resultant ELBOs compare, as well as comparisons of ELBOs and hyperparameters to exact GPs on smaller datasets where feasible (I think that about 15-20k datapoints should be feasible given the compute the authors had access to). Indeed, a lot of the claims throughout the paper center around comparing IterGP to SVGP/SGPR with respect to the exact GP, and so it seems a bit strange that there aren't really any experimental comparisons with respect to exact GPR!

**Questions:**

In order of relative importance:
1. Following from my main weaknesses, could you confirm whether you used the default GPyTorch settings in terms of faster computations? Apologies again if I missed this!
2. Could you confirm why SGPR was not used for Road and Power? My understanding is that the memory cost, which would be the main limitation, should be roughly on the same order as for IterGP-Opt, or am I incorrect?
3. In Fig. 1, could you confirm how you treated the inducing point locations? Although it shows your point nicely, I find it a bit misleading, as in my experience SVGP would never really optimize its inducing locations to be in a (relatively) large gap between datapoints, and so I was surprised to see an inducing location at about 0.6 on the x-axis.
4. Out of interest, since the ELBO for IterGP-Opt is not stochastic, could L-BFGS-B be used there to speed up optimization?
5. It's not clear to me how the statement in lines 111-113 would be implied by the equation above it - could you elaborate please?

Below I'll list some minor questions and typos:
- Could you please change the colors used in the experiment plots and tables? I am unfortunately red-green colorblind, and I found it very difficult to disentangle the colors between SVGP and SGPR and again between IterGP-Opt and -CG. One example of colorblind-friendly colors in matplotlib can be `plt.style.use('tableau-colorblind10')`. Thanks!
- Missing space in line 49 in "introduced,a"
- Admittedly really nitpicky, but could you reframe Eq. 3 and so forth as a maximization problem, or be more clear that you're minimizing the negative ELBO? I think it makes more sense to keep the ELBO as a lower bound to the log marginal likelihood, and so maximizing makes more sense
- line 99: "we optimizer" -> "we optimize"

**Limitations:**

In my opinion the authors have fairly assessed the limitations of their work, namely the inability to do minibatching and the method's limitation to non-conjugate likelihoods.

-----
Post-discussion update
-----
I would like to thank the authors for an engaging discussion. Overall, I was pleased with the quality of the responses and the additional experiments provided. While the work is perhaps not the most ''exciting'' in terms of its empirical results and it has some limitations (e.g., not being able to be minibatched, as pointed out by another reviewer), I don't really think it's useful to judge a paper along these lines, and even so I think this paper represents a solid contribution and has the potential to pave the way for important developments further down the line. Therefore, I am raising my score accordingly.

I would also like the opportunity to address the last comments left by the authors. As I was not able to respond to these before the author-reviewer discussion period ended, I have not taken these into account for my final score. However, I hope that the authors will take them into consideration when they update their paper.

> As we write in the caption of Table 1 in the original submission, the reported number of epochs is determined as follows: "[...] We report the epoch where each method obtained the lowest NLL, and all performance metrics (NLL, RMSE, and wall-clock time) are measured at this epoch. [...]". The rebuttal PDF did not contain this full caption due to the limited space available, but the final version will, of course. The total number of epochs was determined by resource constraints (1000 epochs for Adam, 100 for LBFGS, except for Power; see also lines 235-238 and our rebuttal response). As Figure R1 (rebuttal pdf) shows this results in roughly similar overall training time between for example SGPR trained with LBFGS and IterGP-Opt trained with Adam. We would recommend interpreting Table (R)1 and Figure (R)1 together to understand the generalization performance.

Thanks for your clarification here. Perhaps I have misunderstood which NLL you are looking at, but I am concerned that this amounts to "cheating" - as I understand it you are looking at test metrics to determine when to stop the optimization process, which of course you wouldn't have access to in a real setting. It would be better to either fix the number of epochs or determine some form of early stopping that only relies on training or validation data.

> We would like to politely disagree with the characterization that the choice of optimizer and its parameters doesn't determine the final result (even in theory). The log-marginal likelihood (also the ELBO) as a function of the kernel hyperparameters is generally non-convex and often has multiple (local) minima (for an example see Figure 5.5 of Rasmussen and Williams (2008)). Which modes an optimizer converges to depends on the initialization, the choice of optimizer and the learning rate (schedule / line search).

Yes, I am in agreement here - I had meant my comment in a highly idealized sense where we could assume that different optimizers would reach the same optimum, but it was sloppy at best and likely unrealistic in many settings.

> The reason we would still recommend using Adam over L-BFGS for IterGP-Opt is the increased resource usage. While the performance is slightly better on both datasets as you correctly point out, the runtime is also larger (e.g. Parkinsons ~=1.3x, Bike 1.8x longer) and the memory usage is higher. For IterGP-Opt one optimizes the kernel hyperparameters and the action entries, which in our case amounts to parameters. As, for example, the PyTorch documentation on L-BFGS points out, L-BFGS requires param_bytes * (history_size+1) amount of memory, which for a typical history size (e.g. PyTorch's default of 100) adds significant overhead, as compared to a single gradient, which only requires param_bytes.

I'm ok with this logic favoring Adam over L-BFGS for memory reasons, as long as it is clearly stated. But I do think it's a bit hard to say that Adam is necessarily faster for IterOpt-GP - it seems fairly close, and in my experience (admittedly, with SGPR) L-BFGS has a tendency to optimize much quicker initially but then spend a lot of time searching for the optimum. It might be instructive to show the training curves as well for both optimizers, particularly as the early stopping I've mentioned above might confound these determinations.

Finally, I did want to comment on something I didn't pick up on from a previous response you gave but caught during the reviewer discussion phase:
> As per the recommendation in Section 4.1 of Ober et al (2022), in our original experiments we used a (single precision) Cholesky decomposition both for SGPR and SVGP

Where in Ober et al (2022) does it say they used single precision for SGPR and SVGP? I could not find any mention of single vs. double precision, and I believe it's fairly common for SGPR and SVGP to still be done in double precision, despite the default single precision in GPyTorch.

---

> ### Author Rebuttal · Authors · 2024-08-06
>
> ### Summary
>
> Thank you for your feedback on our paper! We appreciate your detailed recommendations for improvement. Based on these we've added a new set of experiments where we trained SGPR and an exact (Cholesky)GP with LBFGS in double precision (see the rebuttal PDF).
> We believe we have addressed all of your questions fully below, but should anything remain unclear, we are happy to respond during the discussion period.
> Should our response have influenced your initial evaluation of our work, we would be highly appreciative if you would consider updating your score.
> Thank you!
>
> ### Weaknesses
>
> > My main concerns with the paper concern its in its evaluation, particularly with regards to the sparse variational methods. I do not recall seeing these problems on this dataset, although I have admittedly found Parkinson's more problematic, but not in the way shown (I usually find that it just fails with a bad Cholesky decomposition instead of showing divergent behavior). In my experience, this is because while GPyTorch is a state-of-the-art package for conjugate gradient-based methods, it is problematic when it comes to its implementations of sparse variational methods. I believe these instabilities are rather due to GPyTorch's use of Lanczos and conjugate gradient algorithms when computing covariance matrix decompositions and log probabilities, in addition to the use of float32. In my experience, stochastic variational methods are very sensitive to numerical accuracy and so should only be computed using the full Cholesky decomposition and float64. Therefore, since I did not see mention of this (although apologies if I have missed it!) I would recommend changing the GPyTorch defaults to use Cholesky and `float64`, or using a package that does not rely on approximate matrix decompositions such as GPflow or GPJax.
>
> We would like to clarify that we closely followed the recommendations of Ober et al. (2022) and computed any (covariance) matrix decompositions for SGPR and SVGP with Cholesky using an iteratively increasing additive jitter should the decomposition fail. We did *not* use Lanczos or CG.
> The iteratively added jitter could explain the observed divergent behavior instead of an outright failure on Parkinson's.
> We would like to politely point out that GPyTorch does not use Lanczos or CG for variational methods, but rather a Cholesky decomposition (see the source code for [SGPR](https://github.com/cornellius-gp/gpytorch/blob/283105b4f71b0e97fb2dbbf34c95396c7a88bd25/gpytorch/kernels/inducing_point_kernel.py#L65) and [SVGP](https://github.com/cornellius-gp/gpytorch/blob/283105b4f71b0e97fb2dbbf34c95396c7a88bd25/gpytorch/variational/variational_strategy.py#L203)).
> We will expand our description of the experiments with this additional information.
> To alleviate concerns about the use of single precision, we conducted new experiments in double precision as described in the answer to the next question.
>
> > Moreover, one of the key attributes of SGPR is that because the loss is not stochastic, it can be optimized (for both hyperparameters and inducing locations) using second-order optimizers such as L-BFGS-B, as is commonly done. This is typically much faster than Adam, and essentially allows optimization to be parameter-free (as the usual defaults generally work well). It would have been good to compare to this, as is common in the literature, although I do commend the parameter search that the authors did for Adam optimization.
>
> Thank you for this suggestion! We've made the following improvements to our main experiment.
> We trained SGPR on all datasets in double precision using L-BFGS with a Wolfe line search for 100 epochs, based on the recommendations of Ober et al. (2022). We performed a sweep over the initial learning rate $\text{lr}_{\text{SGPR}} \in \\{10^{-4}, 10^{-3}, 10^{-2}, 10^{-1}, 1\\}$ and repeated each run for five random seeds. Both in our original experiments using single precision and Adam and the new ones, we computed Cholesky decompositions with adaptively increasing kernel jitter as per Section 4.1 of Ober et al (2022).
> We find that this indeed increases its performance as the updated Table R1 in the attached rebuttal PDF shows. SGPR now outperforms SVGP on all small to medium datasets as expected, but IterGP-Opt still matches or outperforms SGPR on all datasets, except "Bike", as both Table R1 and Figure R1 in the attached rebuttal PDF show.
>
> > Finally, it would be interesting to see how the resultant ELBOs compare, as well as comparisons of ELBOs and hyperparameters to exact GPs on smaller datasets where feasible (I think that about 15-20k datapoints should be feasible given the compute the authors had access to). Indeed, a lot of the claims throughout the paper center around comparing IterGP to SVGP/SGPR with respect to the exact GP, and so it seems a bit strange that there aren't really any experimental comparisons with respect to exact GPR!
>
> Based on your suggestion we've added results for exact (Cholesky)GPs to our experiments (see Table R1 of the rebuttal PDF). We trained a CholeskyGP on Parkinson's and Bike with LBFGS (using a Wolfe line search) in double precision for 100 epochs. We performed a parameter sweep over the initial learning rate $\mathrm{lr}_{\text{CholeskyGP}} \in \\{10^{-4}, 10^{-3}, 10^{-2}, 10^{-1}, 1\\}$ and repeated each run five times for different seeds.
> Perhaps unsurprisingly, the exact CholeskyGP outperforms all approximations on those small datasets.
> We also added the CholeskyGP to the hyperparameter plot in Figure S4 of the supplementary of our paper. The exact GP tends to learn a larger outputscale and smaller observation noise than the approximate methods. Unfortunately, we could not add this to the rebuttal pdf due to the space constraint of one page only.

---

> > ### Comment · Reviewer_wkY4 · 2024-08-12
> > **Response to rebuttal**
> >
> > Thank you for your rebuttal and detailed response to my review and others. I can confirm that I have carefully read the rebuttals as well as the discussion (up to this point). In general, I think the updated results represent a significant improvement to the quality of the paper. I would also like to thank the authors for correcting some of the confusion I had regarding their paper as well as GPyTorch. However, I generally like to see how the discussion period plays out, and especially if the other reviewers are satisfied with the authors' responses, before committing to change my score one way or another.
> >
> > A couple of minor points still come to mind:
> > - I am somewhat confused by how the number of epochs is chosen for Adam in the updated results. At the end of the day, the results should be the same (theoretically speaking, of course) regardless of how you choose the optimizer, with the only difference really being time taken and memory allocation, but this is not reflected in these updated results. This is especially striking in Parkinsons and Bike for both SGPR and IterGP-Opt, which do not seem to reflect the authors' recommendation for Adam over L-BFGS (as stated in a response to reviewer 7YnY). How did the authors determine when to stop Adam optimization (looking at the paper I couldn't find where this was mentioned, but apologies if I've missed it)?
> > - I appreciate that the authors have agreed to include an analysis of the hyperparameter values of CholeskyGP/exact GPR in their manuscript, but I still think that explicitly considering the ELBOs/LML estimates could be somewhat instructive, as indicated in my original review. Firstly, because solely considering predictive metrics can be misleading when the model is mis-specified, as it most likely is here on these UCI datasets, but also in that it might shed light on some of the discussions being had about optimization.
> >
> > Overall though, while I continue to reserve judgement about the final score until later, I do think that in my view this paper has already been significantly improved by the responses the authors have made.

---

> ### Author Response · Authors · 2024-08-06
> **Answers to Questions**
>
> ### Questions
>
> > 1. Following from my main weaknesses, could you confirm whether you used the default GPyTorch settings in terms of faster computations? Apologies again if I missed this!
>
> As per the recommendation in Section 4.1 of Ober et al (2022), in our original experiments we used a (single precision) Cholesky decomposition both for SGPR and SVGP to compute solves with $K(Z, Z)$ with an iteratively increasing jitter should the decomposition fail. In other words, we did not rely on iterative methods for accelerated solves for SGPR and SVGP. We will make this more explicit in the description of the experimental setup in the final version of the paper.
>
> > 2. Could you confirm why SGPR was not used for Road and Power? My understanding is that the memory cost, which would be the main limitation, should be roughly on the same order as for IterGP-Opt, or am I incorrect?
>
> The asymptotic memory complexity $O(nm)$ is indeed the same for $m=i$, but since we are using $m=1024$ inducing points versus $i=512$ for IterGP-Opt the memory cost of SGPR is two times higher, ignoring any constant factors arising from implementation differences.
> When training SGPR in double precision (with LBFGS), the memory required increases by another factor of two, giving an overall memory requirement that is at least four times higher. These differences lead to prohibitive memory consumption when we attempt to train SGPR on Road on the GPU we used in our experiments.
>
> > 3. In Fig. 1, could you confirm how you treated the inducing point locations? Although it shows your point nicely, I find it a bit misleading, as in my experience SVGP would never really optimize its inducing locations to be in a (relatively) large gap between datapoints, and so I was surprised to see an inducing location at about 0.6 on the x-axis.
>
> We optimized the hyperparameters and inducing points of SVGP in Figure 1 using Adam with an initial learning rate of 0.1 for 200 epochs and a linear learning rate decay. Your experience matches ours that in one dimension the depicted scenario does not occur frequently -- it depends on the initialization and choice of optimization hyperparameters. However, we believe *it faithfully reflects SVGP's behavior in higher dimensions*.
>
> To substantiate this claim, we ran an experiment on synthetic data based on a latent function drawn from a GP with increasing dimension of the input space. We find that the average Mahanalobis distance (induced by the lengthscales) across inducing points to the nearest datapoint increases with dimension (see Figure R2 of author response PDF). We compare this to randomly placed inducing points. While SVGP does move inducing points closer to data points on average, they are still becoming increasingly distant from the data measured in lengthscale units. Further, we also provide more evidence for our claim that SVGP's latent variance often is very small (at inducing points) relative to the total variance, meaning almost all deviation from the posterior mean is considered to be observational noise. The right plot in Figure R2 of the attached rebuttal PDF shows this across dimensions. For $d\geq 4$ approximately 95% of the predictive variance is due to observational noise.
>
> > 4. Out of interest, since the ELBO for IterGP-Opt is not stochastic, could L-BFGS-B be used there to speed up optimization?
>
> Thank you for this suggestion! We ran an additional set of experiments on all but the two largest datasets for IterGP-Opt. We trained using L-BFGS in double precision with a Wolfe line search run for a maximum of 100 epochs and averaged across five seeds. As the results in Table R1 of the attached rebuttal PDF show, performance does increase slightly at comparable runtime (only for Bike the runtime increased considerably).

---

> ### Author Response · Authors · 2024-08-06
> **Answers to Questions [continued]**
>
> > 5. It's not clear to me how the statement in lines 111-113 would be implied by the equation above it - could you elaborate please?
>
> We intended to reference the way the approximate posterior is defined (i.e. Equation (5) and line 105) rather than the equation above line 111 in our statement. Equation (5) and line 105 imply a monotonic decrease in the marginal variance as a function of $i$. Observe that the precision matrix approximation $C_i$ (defined in line 105) has rank $i$, assuming the actions, i.e. columns of $S_i$, are linearly independent and non-zero (as assumed in Wenger et al. (2022)). Therefore we can rewrite
> $C_i = \sum_{\ell=1}^i \tilde{d}\_\ell \tilde{d}_\ell^\top $ as a sum of rank 1 matrices. Note that this is actually how Wenger et al. (2022) compute $C_i$ in Algorithm 1 in their paper. Therefore the marginal variance at iteration $i$ for $i \leq j$ is given by
>
> $$K_{i}(x, x) = K(x, x) - \sum_{\ell=1}^i K(x, X) \tilde{d}\_\ell \tilde{d}\_\ell^\top K(X, x)=K(x, x) - \sum_{\ell=1}^i (K(x, X) \tilde{d}\_\ell)^2 \geq K(x, x) - \sum_{\ell=1}^j (K(x, X) \tilde{d}\_\ell)^2= K_{j}(x,x)$$
>
> Now for $i=n$, since the actions are assumed to be linearly independent and non-zero (see Wenger et al. (2022)), $S_i$ has rank $n$ and thus $C_i = S_i(S_i^\top \hat{K}S_i)^{-1}S_i^\top = \hat{K}^{-1}$. Therefore we have $K_{i}(x, x) \geq K_{j}(x, x) \geq K_{\star}(x, x)$.
>
> > Could you please change the colors used in the experiment plots and tables? I am unfortunately red-green colorblind, and I found it very difficult to disentangle the colors between SVGP and SGPR and again between IterGP-Opt and -CG. One example of colorblind-friendly colors in matplotlib can be plt.style.use('tableau-colorblind10'). Thanks!
>
> We apologize for not taking this into account. We have swapped out the colormap for [Seaborn's colorblind palette `sns.color_palette("colorblind", 5)`](https://seaborn.pydata.org/generated/seaborn.color_palette.html).

---

> ### Author Response · Authors · 2024-08-12
>
> Thank you for carefully reading our responses and of course for your detailed and actionable feedback!
>
> > I am somewhat confused by how the number of epochs is chosen for Adam in the updated results. [...] How did the authors determine when to stop Adam optimization (looking at the paper I couldn't find where this was mentioned, but apologies if I've missed it)?
>
> As we write in the caption of Table 1 in the original submission, the reported number of epochs is determined as follows: "[...] We report the epoch where each method obtained the lowest NLL, and all performance metrics (NLL, RMSE, and wall-clock time) are measured at this epoch. [...]". The rebuttal PDF did not contain this full caption due to the limited space available, but the final version will, of course.
> The *total* number of epochs was determined by resource constraints (1000 epochs for Adam, 100 for LBFGS, except for Power; see also lines 235-238 and our rebuttal response). As Figure R1 (rebuttal pdf) shows this results in roughly similar *overall* training time between for example SGPR trained with LBFGS and IterGP-Opt trained with Adam.
> We would recommend interpreting Table (R)1 and Figure (R)1 together to understand the generalization performance.
>
> > At the end of the day, the results should be the same (theoretically speaking, of course) regardless of how you choose the optimizer, with the only difference really being time taken and memory allocation, [...]
>
> We would like to politely disagree with the characterization that the choice of optimizer and its parameters doesn't determine the final result (even in theory). The log-marginal likelihood (also the ELBO) as a function of the kernel hyperparameters is generally non-convex and often has multiple (local) minima (for an example see Figure 5.5 of [Rasmussen and Williams (2008)](http://gaussianprocess.org/gpml/chapters/RW.pdf)). Which modes an optimizer converges to depends on the initialization, the choice of optimizer and the learning rate (schedule / line search).
>
> > [...] but this is not reflected in these updated results. This is especially striking in Parkinsons and Bike for both SGPR and IterGP-Opt, which do not seem to reflect the authors' recommendation for Adam over L-BFGS (as stated in a response to reviewer 7YnY).
>
> The reason we would still recommend using Adam over L-BFGS for IterGP-Opt is the increased resource usage. While the performance is slightly better on both datasets as you correctly point out, the runtime is also larger (e.g. Parkinsons ~=1.3x, Bike 1.8x longer) and the memory usage is higher. For IterGP-Opt one optimizes the kernel hyperparameters and the action entries, which in our case amounts to $d+2 + n$ parameters. As, for example, the [PyTorch documentation on L-BFGS](https://pytorch.org/docs/stable/generated/torch.optim.LBFGS.html) points out, L-BFGS requires `param_bytes * (history_size+1)` amount of memory, which for a typical history size (e.g. PyTorch's default of 100) adds significant overhead, as compared to a single gradient, which only requires `param_bytes`.
>
> ---
>
> > I appreciate that the authors have agreed to include an analysis of the hyperparameter values of CholeskyGP/exact GPR in their manuscript, but I still think that explicitly considering the ELBOs/LML estimates could be somewhat instructive, as indicated in my original review. Firstly, because solely considering predictive metrics can be misleading when the model is mis-specified, as it most likely is here on these UCI datasets, but also in that it might shed light on some of the discussions being had about optimization.
>
> We apologize for overlooking this suggestion for improvement in your original response! We logged training losses (i.e. ELBOs / (approximate) LMLs) for each epoch in all our runs and will gladly add these to the manuscript (e.g. as an additional column in Table 1, and as an additional row in Figure 1).
>
> ---
>
> We hope our additional answers were helpful in forming your final opinion about our work! If there are any outstanding questions, please do not hesitate to voice them.

---

### Official Review · Reviewer_LXxc · 2024-06-27

**Soundness:** 3
**Presentation:** 4
**Contribution:** 4
**Rating:** 7
**Confidence:** 5

**Summary:**

The authors cast the IterGP method of Wenger et al, which is guaranteed to give conservative estimates of uncertainty relative to the posterior, as a variational procedure. This allows for hyperparameter selection and selection of parameters controlling the quality of the approximation ($\mathbf{S}$) through maximization of an evidence lower bound. The authors compare this method to existing variational approximations on several benchmark datasets.

**Strengths:**

- The exposition is clear and well motivated.
- Placing IterGP within a variational context both allows for a conceptual connection to SGPR/SVGP and (more importantly) practical progress in terms of selection of parameters.

**Weaknesses:**

- Datasets have not been properly cited. In particular, a general citation to UCI doesn’t give credit to the creators of each dataset. License information was also not stated.
- I found the discussion of information gain not well-motivated. The definition of information gain was assumed to be known, and it wasn’t clear from the text why maximizing this quantity would lead to a good quality approximation. If the only goal is to suggest projection onto the largest $i$-eigenvalues, there are many results in the matrix approximation literature that also motivate this approach. Moreover, there is prior work in the Nystrom approximation literature considering the top $i$-eigenvalues as a subspace for performing inference, for example Ferrari-Trecate et al, https://proceedings.neurips.cc/paper/1998/hash/55c567fd4395ecef6d936cf77b8d5b2b-Abstract.html.
- The phrasing in the caption of figure 4 conflates credible and confidence intervals. The accuracy of the credible interval should be with reference to the (exact) posterior and cannot be shown by frequentist coverage. To be clear, I think checking the empirical coverage of the methods is reasonable, but don’t think you can draw conclusions about the accuracy of the credible interval from this.

**Questions:**

- The authors claim SVGP has “no latent uncertainty” where data is absent. Do the authors mean that the posterior variance is actually 0 or extremely close to it away from the data? I don’t see how this can be true since SVGP recovers the posterior distribution in certain limits. If what is meant is that uncertainty is underestimated away from the data, this is quite a bit weaker as a statement and should be made clear.
- “Unlike these other methods, the IterGP posterior updates are defined by a linear combination of kernel functions on training data, which guarantees that posterior variance estimates always over estimate the exact GP uncertainty” — Could you clarify this sentence? First, what is meant by “posterior updates”? Posterior mean updates? Posterior variance updates? Posterior sample path updates? Also, I don’t think this is the property that guarantees that posterior variances are overestimated. It is easy to write down a form of posterior variance that is both a linear combination of kernel function and underestimates the posterior variance, for example 0.
- How does the expression for the approximate predictive standard deviation (section 2) in terms of a worst case quantity over an RKHS ball compare to the analogous statement for low-rank variational methods (given in Theorem 6 of Wild et al 2023, https://arxiv.org/pdf/2106.01121)? I think a brief comparison (possibly in the appendix if space does not allow) would be useful.
- When stating what is meant by computation aware, should there be a convergence condition? I.e. not only should the precision increase, but it should converge to the posterior precision as the amount of compute increases under some sensible restrictions.
- I don’t understand what is plotted on the right of figure 2, or what my takeaway from it should be. What is meant by plotting the magnitude of the eigenvalues of the matrix for each $x_j$? On the plot on the right, what notion of distance between subspaces is used?
- Should I interpret equation 12 as stating the $\mathbf{S}$ is block diagonal? If so, is this said explicitly somewhere?
- Could you explain in more detail why SGPR diverges in these experiments? Have you checked if this is resolved by double precision?


### Other

- Line 49, missing space after comma before “a”
- Line 61, the authors claim that IterGP has “calibrated” confidence. Is the claim that the confidence intervals are calibrated, or that they are conservative in the sense of overestimating the posterior credible intervals?
- Line 73, h is introduced here as the latent function and never appears again. I’m a bit unclear why it is needed.
- Line 82, before discussing computation in model selection, it should be stated that you assume maximum (marginal) likelihood will be used. There are alternative approaches (e.g. sampling)

**Limitations:**

The authors point out limitations well; most importantly, the restriction to Gaussian likelihoods in the present work.

---

> ### Author Rebuttal · Authors · 2024-08-06
>
> ### Summary
>
> Thank you for your positive review of our paper and helpful input!
> We responded to your questions below. In particular, we've reformulated the motivation behind choosing actions according to an information-theoretic objective and we've added a statement proving the monotonic decrease in marginal variance to the exact posterior marginal variance as a function of $i$.
> If anything remains unclear, we are happy to elaborate further during the discussion period!
>
> ### Weaknesses
>
> > - Datasets have not been properly cited. In particular, a general citation to UCI doesn’t give credit to the creators of each dataset. License information was also not stated.
>
> We have added citations for each of the datasets we have used and added license information to the supplementary.
>
> > - I found the discussion of information gain not well-motivated. The definition of information gain was assumed to be known, and it wasn’t clear from the text why maximizing this quantity would lead to a good quality approximation. If the only goal is to suggest projection onto the largest $i$-eigenvalues, there are many results in the matrix approximation literature that also motivate this approach. Moreover, there is prior work in the Nyström approximation literature considering the top $i$-eigenvalues as a subspace for performing inference, for example Ferrari-Trecate et al, https://proceedings.neurips.cc/paper/1998/hash/55c567fd4395ecef6d936cf77b8d5b2b-Abstract.html.
>
> Based on your feedback, we've reformulated the information-theoretic motivation for projecting onto the top $i$-eigenvalue subspace as follows. We've replaced Lemma S2 with an arguably more intuitive result, namely that an "optimal" linear combination of the data should maximally reduce our uncertainty about the latent function at the training data or equivalently maximally increase the divergence of our updated belief from the prior. More formally, it holds that the actions maximizing the information gain, defined as the difference in entropy between prior and posterior, i.e.
>
> $$
> S_i = \text{argmax}_{S \in \mathbb{R}^{n \times i}} ( H(f(X)) - H(f(X) \mid S^{\top} y) ) = \text{argmax}\_{S \in \mathbb{R}^{n \times i}} \text{KL}(p(f \mid S^\top y)\ || \ p(f))
> $$
>
> are given by the top-$i$ eigenvectors of $\hat{K}$, where $y \in \mathbb{R}^n$. This is different from the previous formulation in equation (10) in that here we are directly reasoning about the latent function of interest, $f$.
> In Bayesian information-theoretic formulations of active learning, choosing observations that minimize posterior entropy (or a myopic approximation defined by the expected information gain) has a long history (e.g. MacKay (1992), Section 2 of Houlsby et al. (2011)). Here, rather than choosing observations directly, we generalize this idea to choosing *linear combinations of observations*.
> We've included this improved result and its motivation in the final version.
> We've also added a citation to the work by Ferrari-Trecate to make the connection to prior work explicit.
>
> - MacKay, D. J. C. (1992). Information-Based Objective Functions for Active Data Selection. Neural Computation, 4(4), 590–604. https://doi.org/10.1162/neco.1992.4.4.590
> - Houlsby, N., Huszár, F., Ghahramani, Z., & Lengyel, M. (2011). Bayesian Active Learning for Classification and Preference Learning. arXiv. https://doi.org/10.48550/arXiv.1112.5745
>
> > - The phrasing in the caption of Figure 4 conflates credible and confidence intervals. The accuracy of the credible interval should be with reference to the (exact) posterior and cannot be shown by frequentist coverage. To be clear, I think checking the empirical coverage of the methods is reasonable, but don’t think you can draw conclusions about the accuracy of the credible interval from this.
>
> Thank you for pointing this out to us. We have modified the caption to describe that we are measuring the empirical coverage of the credible interval of IterGP-Opt and SVGP and have removed any claim about the accuracy of the credible intervals.

---

> > ### Author Response · Authors · 2024-08-07
> >
> > We would like to issue a brief correction here to one equation in our response above. The second equality in the definition of the information-optimal actions should be the entropy of the computation-aware posterior, rather than the KL divergence to the prior:
> >
> > $$
> > S_i = \dots = \textrm{argmin}_{S \in \mathbb{R}^{n \times i}} \mathbb{E}\_{p(f(X) \mid S^\top y)}[- \log p(f(X) \mid S^\top y)]
> > $$

---

> > > ### Comment · Reviewer_LXxc · 2024-08-12
> > >
> > > The authors have adressed the concerns I raised in my review. I maintain my initial score and assessment that the paper should be accepted.

---

> ### Author Response · Authors · 2024-08-06
> **Answers to Questions**
>
> ### Questions
>
> > - The authors claim SVGP has “no latent uncertainty” where data is absent. Do the authors mean that the posterior variance is actually 0 or extremely close to it away from the data? I don’t see how this can be true since SVGP recovers the posterior distribution in certain limits. If what is meant is that uncertainty is underestimated away from the data, this is quite a bit weaker as a statement and should be made clear.
>
> We would like to clarify that we claim this in reference to Figure 1 as an illustration of a more general phenomenon that can occur when using SVGP. Specifically, we claim that "SVGP, [...] is overconfident at the locations of the inducing points if they are not in close proximity to training data, which becomes increasingly likely in higher dimensions. This phenomenon can be seen in a toy example in Figure 1 (middle left), where SVGP has no latent uncertainty where data is absent (lower row, cf. other methods)."
> We've modified this claim to "has near zero latent uncertainty at the inducing point away from the data".
> Additionally in the caption of Figure 1 we write referring to the same figure, that "SVGP expresses almost no posterior variance in the middle region, and thus almost all deviation from the posterior mean is considered to be observational noise."
>
> To back up these two claims we added a new experiment, which measures the distance of inducing points to the data (measured in lengthscale units) as a function of input dimension and the ratio between posterior and predictive variance at the inducing points (see Figure R2 of rebuttal PDF). We find that inducing points are optimized to lie increasingly farther away from the data as the dimension of the input space increases. Further, we observe that in higher dimensions ($d\geq 4$) the latent variance can be as little as 5% of the predictive variance.
> Note that all our statements and observations are made under the assumption of a fixed number of data points $n > m$ and inducing points $m$ and therefore do not conflict with statements about the convergence of SVGP to the exact posterior.
>
> > - “Unlike these other methods, the IterGP posterior updates are defined by a linear combination of kernel functions on training data, which guarantees that posterior variance estimates always over estimate the exact GP uncertainty” — Could you clarify this sentence? First, what is meant by “posterior updates”? Posterior mean updates? Posterior variance updates? Posterior sample path updates? Also, I don’t think this is the property that guarantees that posterior variances are overestimated. It is easy to write down a form of posterior variance that is both a linear combination of kernel function and underestimates the posterior variance, for example 0.
>
> We refer to the posterior variance updates and specifically *the form of the downdate*. We've clarified this in the draft. To see why this guarantees the variances are always overestimated (they monotonically decrease as a function of $i$), consider the following. Observe that the precision matrix approximation $C_i$ (defined in line 105) has rank $i$, assuming the actions, i.e. columns of $S_i$, are linearly independent and non-zero (as assumed in Wenger et al. (2022)). Therefore we can rewrite
> $C_i = \sum_{\ell=1}^i \tilde{d}\_\ell \tilde{d}_\ell^\top $ as a sum of rank 1 matrices. Note that this is actually how Wenger et al. (2022) compute $C_i$ in Algorithm 1 in their paper. Therefore the marginal variance at iteration $i$ for $i \leq j$ is given by
>
> $$K_{i}(x, x) = K(x, x) - \sum_{\ell=1}^i K(x, X) \tilde{d}\_\ell \tilde{d}\_\ell^\top K(X, x)=K(x, x) - \sum_{\ell=1}^i (K(x, X) \tilde{d}\_\ell)^2 \geq K(x, x) - \sum_{\ell=1}^j (K(x, X) \tilde{d}\_\ell)^2= K_{j}(x,x)$$
> Now for $i=n$, since the actions are assumed to be linearly independent and non-zero, $S_i$ has rank $n$ and thus $C_i = S_i(S_i^\top \hat{K}S_i)^{-1}S_i^\top = \hat{K}^{-1}$. Therefore we have $K_{i}(x, x) \geq K_{j}(x, x) \geq K_{\star}(x, x)$.

---

> > ### Author Response · Authors · 2024-08-06
> > **Answers to Questions [continued]**
> >
> > > - When stating what is meant by computation-aware, should there be a convergence condition? I.e. not only should the precision increase, but it should converge to the posterior precision as the amount of compute increases under some sensible restrictions.
> >
> > In an earlier answer, we demonstrated that the marginal variance decreases monotonically as a function of $i$ (albeit not necessarily strictly), given linearly independent, non-zero actions. We immediately obtain that the marginal precision increases monotonically up to the marginal precision of the posterior as the computational budget increases, i.e. $i \to n$.
> >
> > > - I don’t understand what is plotted on the right of figure 2, or what my takeaway from it should be. What is meant by plotting the magnitude of the eigenvalues of the matrix for each $x_j$? On the plot on the right, what notion of distance between subspaces is used?
> >
> > One way to understand how different instances of IterGP differ based on different choices of actions, i.e. the columns of the matrix $S_i$, is by interpreting the magnitude of the action entries as how the computational budget is distributed across the $n$ observations (see also Figure 2 of Wenger et al. (2022)). This interpretation arises from the formulation of IterGP conditioning on $S_i^\top y$ (see also lines 122-125). For example, unit vector actions correspond to spending all budget on a single data point, sparse actions target only a subset of data points, and CG and eigenvector actions distribute the budget across all data points with different weights. The top row in Figure 2 corresponds to eigenvector actions, hence we plot the magnitude of the eigen*vector* entries. The rows below illustrate the two variants of IterGP which we consider in our paper. In the plot on the right in Figure 2, we show the Grassman distance, which is the Euclidean norm of the vector of principal angles between the two subspaces. We've added its definition to the paper.
> >
> > > - Should I interpret equation 12 as stating the $S$ is block diagonal? If so, is this said explicitly somewhere?
> >
> > The matrix $S$ consists of $i$ blocks (*), which are of size $k \times 1$. We state this implicitly in lines 183-184, but we've made this more explicit in the final draft.
> >
> > > - Could you explain in more detail why SGPR diverges in these experiments? Have you checked if this is resolved by double precision?
> >
> > For SGPR, we computed Cholesky decompositions in single precision with an iteratively increasing additive jitter should the decomposition fail as per Section 4.1 of Ober et al (2022).
> > The iteratively added jitter could explain the observed divergent behavior instead of an outright failure in single precision.
> > During the rebuttal, we added results for SGPR trained in double precision with LBFGS to our experiments, which does not show any divergence during training as Figure R1 in the attached rebuttal PDF shows. Training in this way improves the performance of SGPR over SVGP, but IterGP-Opt still matches or outperforms SGPR, except on "Bike".
> >
> > > - Line 61, the authors claim that IterGP has “calibrated” confidence. Is the claim that the confidence intervals are calibrated, or that they are conservative in the sense of overestimating the posterior credible intervals?
> >
> > The claim is that they are conservative. We've corrected this in the final version.

---

### Official Review · Reviewer_7YnY · 2024-07-12

**Soundness:** 3
**Presentation:** 3
**Contribution:** 3
**Rating:** 5
**Confidence:** 4

**Summary:**

This paper presents a substantial extension to an existing class of models called IterGP. By introducing a novel training loss which combines both the hyperparameters and a sparse action matrix, the IterGP-Opt model offers linear-time scalability. Experiments over a range of UCI datasets demonstrate that IterGP-Opt matches or outperforms the benchmark SGPR and SVGP models.

**Strengths:**

The paper is well written and clearly structured.

The proposed IterGP-Opt model displays a high degree of novelty.

The results and methodology are for the most part well presented.

Scalable inference is an area of high impact, so potential improvements over the SVGP model are of high interest.

**Weaknesses:**

I have a couple of concerns regarding the models used in the experiments:

The paper cites Ober et al "Recommendations for Baselines and Benchmarking Approximate Gaussian Processes" as inspiration for including SGPR as a strong benchmark, but does not follow the guidance for SGPR training outlined in that paper. Both in terms of the choice of optimiser, and the approach of not jointly optimising the inducing points with the kernel hyperparameters. This is concerning as it could be having a significantly detrimental impact on the performance of the SGPR baseline - as is further evidenced by it underperforming SVGP on three of the four tasks. In any case, given the popularity of the datasets used, it ought to be straightforward to verify the SGPR performance is consistent with other publications in the literature.

One of the other methods presented in Table 1 is "IterGP-CG", but its status should be clarified for the reader. The first reference to IterGP-CG in the paper is in the caption to Figure 1, where it is introduced as "IterGP-CG (ours), ...", implying this model is a novelty of the paper. Yet later in the Background section, and also in the footnote on page 7, it is pointed out that the IterGP-CG model is not new. Given that IterGP-CG is not mentioned at all in the Conclusion section, it seems likely the "ours" was just a typo, but this is of course a very important point to clarify! Table 1 itself should also be updated to clarify which method(s) are claimed to be novel to the paper.

**Questions:**

The final sentence of the abstract claims "As a result of this work, Gaussian processes can be trained on large-scale datasets without compromising their ability to quantify uncertainty" - Doesn't this imply we are conducting exact inference on these large-scale datasets? I'd recommend the authors consider a suitable rephrasing such as "significantly compromising".

"SGPR typically requires more gradient iterations during training as it introduces new parameters (the inducing point locations Z)."
I was surprised by this comment, because in my experience SGPR typically requires significantly fewer training iterations than SVGP. In part because training is unbatched, allowing it to exploit standard LBFGS optimisation as opposed to using Adam. And also because it is often not desirable to train the inducing point locations jointly with the hyperparameters.

**Limitations:**

The authors include a well written section on the limitations of IterGP-Opt.

---

> ### Author Rebuttal · Authors · 2024-08-06
>
> ### Summary
>
> Thank you for your time and effort in reviewing our paper and in particular for suggesting improvements to our benchmark experiments.
> We have addressed your main concerns with a set of new experiments where we train SGPR using LBFGS in double precision. These changes close the gap between SGPR and SVGP, but IterGP-Opt still matches or outperforms SGPR on all datasets, except "Bike" (see the rebuttal PDF).
> You can find our responses to your questions below. If anything remains unclear, we are happy to respond during the discussion period.
> Should the changes we've made to our paper alleviate the concerns you voiced, we would be grateful if you would update your score to reflect this. Thank you!
>
> ### Weaknesses
>
> > The paper cites Ober et al [...] as inspiration for including SGPR as a strong benchmark, but does not follow the guidance for SGPR training outlined in that paper. [...] This is concerning as it could be having a significantly detrimental impact on the performance of the SGPR baseline - as is further evidenced by it underperforming SVGP on three of the four tasks. In any case, given the popularity of the datasets used, it ought to be straightforward to verify the SGPR performance is consistent with [...] the literature.
>
> To ensure the SGPR baseline is well-tuned and to alleviate your concerns, we made the following improvements to our main experiment:
>
> - We trained SGPR on all datasets in double precision using L-BFGS with a Wolfe line search for 100 epochs in line with the recommendations of Ober et al. (2022). We performed a sweep over the initial learning rate $\text{lr}_{\text{SGPR}} \in \\{10^{-4}, 10^{-3}, 10^{-2}, 10^{-1}, 1\\}$ and repeated each run for five random seeds. Both in our original experiments using single precision and Adam and the new ones, we computed Cholesky decompositions with adaptively increasing kernel jitter as per Section 4.1 of Ober et al (2022).
> We find that this indeed increases its performance. SGPR now outperforms SVGP on all small to medium datasets as expected, but IterGP-Opt still matches or outperforms SGPR on all datasets, except "Bike", as both Table R1 and Figure R1 in the attached rebuttal PDF show. Thank you for suggesting this improvement to the SGPR baseline!
>
> - Additionally, we checked our newly generated results against the corresponding results reported by Ober et al. (2022) in Tables 2 and 3 of Appendix H and found the newly reported performance of SGPR in our experiments to be largely consistent on all datasets in common between their and our work ("Bike", "Protein", "KEGGu") for $M=1000$ inducing points. However, one should be careful when comparing these numbers, since Ober et al. (2022) use a squared exponential kernel, while we use a Matern(3/2) kernel.
>
> - Finally, based on the suggestions of reviewer wkY4, we added an exact (Cholesky)GP baseline on the two smallest datasets and
> also trained IterGP-Opt using L-BFGS. We found that this marginally improves the performance of IterGP-Opt. However, due to the higher memory consumption and slightly larger runtime, we would still recommend using Adam in practice.
>
> > One of the other methods presented in Table 1 is "IterGP-CG", but its status should be clarified for the reader. The first reference to IterGP-CG in the paper is in the caption to Figure 1, where it is introduced as "IterGP-CG (ours), ...", implying this model is a novelty of the paper. Yet later in the Background section, and also in the footnote on page 7, it is pointed out that the IterGP-CG model is not new. Given that IterGP-CG is not mentioned at all in the Conclusion section, it seems likely the "ours" was just a typo, but this is of course a very important point to clarify! Table 1 itself should also be updated to clarify which method(s) are claimed to be novel to the paper.
>
> We apologize for any confusion this typo may have caused.
> As we write in the introduction, IterGP as a class of GP *posterior approximation* methods was introduced by Wenger et al. (2022), one special case being IterGP-CG.
> We did not intend in any way to claim credit for this method.
> The work by Wenger et al. (2022), however, does *not* describe a way to perform *model selection*, which is one of the two main contributions of our work. In particular, we demonstrate in this work how to perform model selection for IterGP-CG. Our second contribution is that we introduce a new variant of IterGP (i.e. IterGP-Opt), which enables posterior approximation in *linear time*. We have modified the draft to more clearly indicate our contributions without inadvertently suggesting we proposed IterGP-CG.
>
>
> ### Questions
>
> > [...] I'd recommend the authors consider a suitable rephrasing such as "significantly compromising" [in the abstract].
>
> Thank you for this feedback. We will adjust the last sentence of the abstract accordingly.
>
> > SGPR typically requires more gradient iterations during training as it introduces new parameters (the inducing point locations $Z$)." I was surprised by this comment, because in my experience SGPR typically requires significantly fewer training iterations than SVGP. [...]
>
> We would like to politely point out that this is a misunderstanding of what we write. *We are comparing SGPR to exact GPs, not to SVGP* in the quoted sentence. We write -- *after* introducing exact GPs and *before* introducing SVGP -- in the paragraph on SGPR in lines 93-95 that "While these complexities significantly improve upon $O(n^3)$ computation/$O(n^2)$ memory, SGPR typically requires more gradient iterations during training as it introduces new parameters (the inducing point locations $Z$)."  We will clarify this in the final version and also add that the inducing point locations can be
> chosen in a way that does not require optimization, e.g. as outlined in Burt et al. (2020) and recommended by Ober et al. (2022).

---

> > ### Comment · Reviewer_7YnY · 2024-08-12
> >
> > Thank you for responding to my concerns, and updating the experimental table accordingly. A few suggestions regarding the table:
> >
> > - The CholeskyGP model is chosen to be light grey, which when bolded barely stands out. Please adjust accordingly, simply making it black should suffice (or alternatively, don't set it to bold at all and only highlight the best of the approximate methods?).
> >
> > - There seem to be learning rates given for the new BFGS-optimised methods, is this just a typo?
> >
> > >  In particular, we demonstrate in this work how to perform model selection for IterGP-CG.
> > - Regarding the presentation of the results for IterGP-CG: in order to make this contribution much clearer, might it be helpful to explicitly compare these results against the previous training procedure for IterGP-CG, as was used in e.g. Figure 4 of arXiv:2205.15449. This would aid the reader to gauge the significance of the contribution.

---

> ### Author Response · Authors · 2024-08-12
>
> > The CholeskyGP model is chosen to be light grey, which when bolded barely stands out. Please adjust accordingly, simply making it black should suffice (or alternatively, don't set it to bold at all and only highlight the best of the approximate methods?).
>
> We are happy to make the CholeskyGP stand out more.
>
> > There seem to be learning rates given for the new BFGS-optimised methods, is this just a typo?
>
> This is not a typo. As we write in the response above this is the *initial* learning rate, which then gets updated by the line search. See also the [corresponding lines](https://github.com/pytorch/pytorch/blob/26b0a0c2f37a8ad376f261df7bb4fee65ff2f230/torch/optim/lbfgs.py#L419-L423) in the PyTorch implementation of L-BFGS and its [`lr` argument](https://pytorch.org/docs/stable/generated/torch.optim.LBFGS.html).
>
> > Regarding the presentation of the results for IterGP-CG: in order to make this contribution much clearer, might it be helpful to explicitly compare these results against the previous training procedure for IterGP-CG, as was used in e.g. Figure 4 of arXiv:2205.15449. This would aid the reader to gauge the significance of the contribution.
>
> We believe there might be a misunderstanding here as to what the contribution of Wenger et al. (2022) is. In Figure 4 they show the loss as a function of the solver iterations when computing an *increasingly better approximate posterior with fixed hyperparameters*. We propose a *model selection / hyperparameter optimization procedure* for this approximate posterior computed with a fixed number of solver iterations (i=512). Wenger et al. (2022) *do not consider model selection / propose a training procedure*. It is therefore not possible to compare their experiment to ours, since the two papers consider two different problems.
>
> Does this clarify our contribution? We are happy to give more detail if that should alleviate your concerns.

---

> > ### Comment · Reviewer_7YnY · 2024-08-12
> >
> > Yes I understand the significance of Figure 4, that during optimisation the hyperparameters were fixed, but there is still a procedure on how to obtain those fixed hyperparameters as cited in the paper:
> > > For all datasets, we select hyperparameters using the training procedure of Wenger et al. [29].
> >
> > The point is, it is important to demonstrate explicitly what improvement is brought to IterGP-CG compared to what was attainable with IterGP-CG before.

---

> ### Author Response · Authors · 2024-08-12
>
> The hyperparameters were chosen with the training procedure for CGGP as proposed in Wenger et al. [29] to favor CGGP. They do the same in the follow-up experiment in Figure 5 where they choose the hyperparameters given by SVGP for their variant IterGP-PI.
>
> Would you be satisfied if we added CGGP training runs and then reported the test performance for the approximate posterior computed by IterGP-CG (as done by Wenger et al 2022)? This would be possible.

---

> > ### Author Response · Authors · 2024-08-12
> >
> > Based on your request, we have added the following **experiment comparing CGGP trained via the procedure given in [Wenger et al. (2022b)](https://arxiv.org/abs/2107.00243) and our proposed model selection procedure for IterGP-CG**:
> >
> > We trained CGGP on all small to medium-sized datasets via L-BFGS for 100 epochs in double precision with a sweep over the initial learning rate. Due to the very limited time until the end of the rebuttal period we ran this experiment only for one seed, we will add the remaining runs for the camera-ready version.
> > As the table below shows, *IterGP-CG trained via our proposed objective outperforms or matches CGGP on all datasets (i.e difference in performance larger than one standard deviation), except in terms of RMSE on Protein.*
> > The significant lower runtime on KEGGu is explained by the fact that after epoch 9 CGGP ceases to improve.
> >
> > | Dataset    | Method    | Optim. | LR  | Epoch | Test NLL | Test RMSE | Avg. Runtime |
> > |------------|-----------|--------|-----|-------|----------|-----------|--------------|
> > | Parkinsons | CGGP      | L-BFGS | 1.0 | 68    | -2.734   | 0.011     | 1min 23s     |
> > |            | IterGP-CG | Adam   | 1.0 | 250   | -2.936   | 0.009     | 1min 44s     |
> > | Bike       | CGGP      | L-BFGS | 1.0 | 15    | -2.053   | 0.021     | 2min 14s     |
> > |            | IterGP-CG | Adam   | 1.0 | 250   | -2.042   | 0.024     | 5min 17s     |
> > | Protein    | CGGP      | L-BFGS | 0.1 | 28    | 0.852    | 0.511     | 16min 13s    |
> > |            | IterGP-CG | Adam   | 1.0 | 27    | 0.820    | 0.542     | 1min 26s     |
> > | KEGGu      | CGGP      | L-BFGS | 1.0 | 9     | -0.510   | 0.127     | 7min 29s     |
> > |            | IterGP-CG | Adam   | 1.0 | 229   | -0.699   | 0.120     | 39min 5s     |
> >
> >
> > Now since CGGP and IterGP-CG share the same posterior mean (see Section 2.1 of [Wenger et al (2022)](http://arxiv.org/abs/2205.15449)), the Test RMSE for IterGP-CG is guaranteed to be identical when using the same hyperparameters. Empirically as Wenger et al (2022) show in Figure 4, the Test NLL is also the same. Therefore the reported numbers here will be the same if computing the approximate posteriors with IterGP-CG for the hyperparameters given by CGGP as done in Wenger et al. (2022). (Note that the reported 2x speedup of IterGP-CG vs CGGP by Wenger et al. 2022 only applies to inference, not the CGGP training procedure).
> >
> > Thank you for your suggestion to improve our paper. We would kindly ask you to take this newly added experiment into account in your final evaluation.

---

> > > ### Comment · Reviewer_7YnY · 2024-08-13
> > >
> > > Thank you for providing the additional experiments, I feel it offers important context, as the reader can now gauge the impact of the contribution (i.e. compared to using sub-optimal hyperparameters for IterGP-CG).
> > > I'm happy to update my score accordingly, and looking forward to the discussion phase.

---

> > > > ### Author Response · Authors · 2024-08-13
> > > >
> > > > Thank you for your suggestions and continuous engagement during the discussion phase!

---

### Author Rebuttal · Authors · 2024-08-06

We would like to thank all reviewers for their time and effort in reviewing our work!
We feel that your helpful suggestions enabled us to improve our paper, in particular, we've added the following additional experiments (see the attached PDF and our individual rebuttals below):

- **Generalization experiments** (Table R1, Figure R1)
    - SGPR trained with LBFGS in double precision on all small and medium sized datasets.
    - IterGP-Opt trained with LBFGS in double precision on all small and medium sized datasets.
    - CholeskyGP baseline trained with LBFGS in double precision on all datasets where this is possible given memory constraints.
- **Experiment on inducing point placement and posterior vs predictive uncertainty of SVGP in higher dimensions** (Figure R2).

We responded to all your questions in individual answers below. If anything remains unclear please let us know!

---

### Decision · Program_Chairs · 2024-09-25

**Decision:**

Accept (poster)

**Comment:**

A summary of the strengths and weaknesses based on the reviews and rebuttal (including the follow-up discussion and that among the reviewers) is provided below:


**STRENGTHS**

1. This paper proposes a novel linear-time IterGP-Opt model by minimizing the (negative) ELBO training loss that optimizes the hyperparameters as well as parameters controlling the approximation quality via variational inference. Such models do not suffer from overconfidence in terms of the uncertainty of their predictions.

2. The paper is well-written.


**WEAKNESS**

1. As the authors have highlighted, the proposed IterGP-Opt is not amenable to stochastic minibatching and hence cannot achieve constant time/memory per iterative step. Nonetheless, the reviewers think that this paper can potentially inspire follow-up works with stochastic minibatching.

2. An initial concern was the unfair comparison with SGPR/DTC as little effort has been put into getting SGPR to perform competitively well. The authors' rebuttal has resolved this issue.

3. The empirical evaluation is lacking. In particular, the baselines are at least a decade old and more powerful derivatives of these baselines and others have since been developed. See, for example, the following work (and references therein) that can also perform hyperparameter optimization/model selection and inference, and the approximation quality can be improved by trading off time efficiency: A Distributed Variational Inference Framework for Unifying Parallel Sparse Gaussian Process Regression Models. ICML 2016.

If not empirically comparing with the above, the authors should at least give a qualitative comparison and its implications on the resulting performance difference.

Despite the weaknesses, we think that the novelty and significance of this work (specifically, point 1 in strength) are sufficiently strong and outweigh such weaknesses.